# What do you know?
# Bayesian knowledge inference for navigating agents

**Matthias Schultheis**
Center for Cognitive Science
Technische Universität Darmstadt
Darmstadt, Germany
matthias.schultheis@tu-darmstadt.de

**Jana-Sophie Schönfeld**
Center for Cognitive Science
Technische Universität Darmstadt
Darmstadt, Germany
janasophie.schoenfeld@stud.tu-darmstadt.de

**Constantin A. Rothkopf**
Center for Cognitive Science
Technische Universität Darmstadt
Darmstadt, Germany
constantin.rothkopf@tu-darmstadt.de

**Heinz Koeppl**
Center for Cognitive Science
Technische Universität Darmstadt
Darmstadt, Germany
heinz.koeppl@tu-darmstadt.de

## Abstract

Human behavior is characterized by continuous learning to reduce uncertainties about the world in pursuit of goals. When trying to understand such behavior from observations, it is essential to account for this adaptive nature and reason about the uncertainties that may have led to seemingly suboptimal decisions. Nevertheless, most inverse approaches to sequential decision-making focus on inferring cost functions underlying stationary behavior or are limited to low-dimensional tasks. In this paper, we address this gap by considering the problem of inferring an agent's knowledge or awareness about the environment based on a given trajectory. We assume that the agent aims to reach a goal in an environment they only partially know, and integrates new information into their plan as they act. We propose a Bayesian approach to infer their latent knowledge state, leveraging an approximate navigation model that optimistically incorporates partial information while accounting for uncertainty. By combining sample-based Bayesian inference with dynamic graph algorithms, we achieve an efficient method for computing posterior beliefs about the agent's knowledge. Empirical validation using simulated behavioral data and human data from an online experiment demonstrates that our model effectively captures human navigation under uncertainty and reveals interpretable insights into their environmental knowledge.

## 1 Introduction

As humans, we navigate a world full of uncertainty. Whether walking through a new city, browsing the internet, or finding the way to the airport, we constantly plan under partial knowledge, balance risks with opportunities, and revise our strategies as we gather new information. This interplay between action and learning is a hallmark of adaptive, intelligent behavior.

Consider the following scenario: you are driving to the airport using your usual route, which you know is typically the fastest. Midway through, you encounter unexpected construction blocking the road. You turn around and take a longer detour. To an external observer unaware of your prior knowledge, your behavior might appear irrational or goal-incongruent—why detour into a neighborhood instead of heading directly to the airport? However, by reasoning about what you likely knew when the decision was made, a more accurate and interpretable explanation emerges: you simply did not

39th Conference on Neural Information Processing Systems (NeurIPS 2025).

know about the construction when you started, and your actions reflect effective planning under that partial knowledge. This example highlights a fundamental challenge in understanding and modeling human behavior: observed decisions are shaped not only by goals and preferences, but also by what they know—and don't know. Nevertheless, most existing inverse sequential decision-making approaches focus on recovering cost functions representing the agent's objectives [1, 2] or assume stationary policies with complete knowledge of the environment structure [3, 4]. More general formulations exist [5, 6], but their computational cost limits them to very low-dimensional domains. These assumptions complicate the applicability to learning agents in general and human behavior in natural tasks in particular, which are characterized by uncertainty, learning, and adaptation.

In this work, we address the problem of inferring an agent's latent knowledge state in a partially known environment based on observed decisions. We focus on navigation tasks where agents act toward a goal, while simultaneously integrating new information about the environment's structure into their plans. From an observed trajectory, our goal is to infer which parts of the world the agent knew beforehand and which were newly incorporated while acting. We assume that, as researchers, we have complete knowledge of the task and environment, but must reason about the subject's latent knowledge state. Although we frame our approach in terms of knowledge of the environment, the same formulation applies to reasoning about an agent's memory or awareness during planning, enabling us, for example, to infer what information subjects transferred between tasks, which parts of the environment were represented in memory, and how awareness influenced planning.

The key challenge of this problem lies in the complexity of reasoning over a large space of possible knowledge states and their corresponding optimal policies. To address this, we combine sample-based Bayesian inference with a tractable navigation model based on dynamic graph algorithms to efficiently compute posterior beliefs over the agent's knowledge state. We evaluate our approach using two simulated tasks and behavioral data from an online human experiment. Our results demonstrate that the model successfully captures human navigation behavior under incomplete information and provides reasonable beliefs of the subjects' prior knowledge in this task, improving in many cases over both an agnostic and heuristic baseline.

## 1.1 Related work

The problem of inferring goals, beliefs, and knowledge when observing trajectories of agents was approached with Bayesian inverse planning [5, 6] in environments with up to 6 belief states. While this method could in principle be applied to our setting of inferring the knowledge of navigating agents, the exact computation of both the posterior belief and the underlying policy is individually intractable in the high-dimensional belief space we consider in our tasks (up to $2^{169}$ belief states). Other work on Bayesian inverse planning used online path planning [7] and adaptive shortest-path algorithms for varying goal states [8, 9]. Rabinowitz et al. [10] introduced with ToMnet a general approach to learn representations of an agent's mind that can be used to predict future behavior, lacking an explicit representation of the agent's knowledge.

In the field of reinforcement learning, inverse approaches for sequential decision-making mostly focus on learning an agent's goals in the form of a reward function and fall under the term inverse reinforcement learning (IRL) [1, 2, 11]. Some of these works also consider the setting of spatial navigation [12]. There is also work to infer internal dynamics models [13–15] and construals [16] of observed agents. Some work on IRL specifically addresses the scenario where agents have uncertainties by regarding POMDPs [17, 3, 4] and can additionally infer variabilities and subjective beliefs [18, 19]. Further, inferring reward functions of learning agents was considered in [20, 21] and inferring their learning rules in [22]. There is also work on inferring an agent's belief about an unknown discount parameter [23] and work on belief inference under incomplete model knowledge for observed fully informed agents [24]. Ashwood et al. [25] considered the problem of estimating reward functions that evolve randomly over time, which shares similarities to our work, as it also deals with high-dimensional inference with non-stationary policies. They deal with the problem by using an expectation-maximization scheme and separate the stationary state-dependent reward component from the time-dependent goal map process. In our case, however, state and time are not separable: the knowledge at a certain time depends on the history of visited states.

For acting under uncertainty, the tradeoff between maximizing reward and gathering information is known as the exploration-exploitation tradeoff [26]. The exploration behavior of humans has been mostly studied in bandit tasks, where actions do not influence the state of the environment [27–30].

In this regard, there is work considering restless bandit tasks [31], spatially correlated bandits [29], and contextual bandits [32]. It was also shown that humans can adapt their exploration behavior to the structure of the environment [33, 34]. Research on how humans form representations for planning and acting when learning the dynamics of the environment was regarded for non-spatial [35] and spatial navigation tasks [36]. Humans' ability to adapt their planning strategy depending on the structure of the problem was investigated in [37]. The variability of humans' planning has been modeled using shortest path algorithms in a softmax policy [38, 8, 39]. Our planning model is conceptually related to the Explicit Explore or Exploit (E$^3$) algorithm [40], which similarly separates known and unknown states. However, while E$^3$ treats unknown states as maximally rewarding to encourage exploration, our model assigns a traversal probability for uncertainty, leading instead to uncertainty aversion. Inferring the goals and plans of artificial agents has been regarded in the field of plan recognition [41–43]. Furthermore, our method can be interpreted as learning properties about the given graph (i.e., the knowledge variables) and is therefore related to graph learning [44], which aims to reconstruct a graph from data. Standard graph learning approaches, however, typically do not capture subjective, temporally updated knowledge states.

## 2 Background

### 2.1 Markov decision processes (MDPs) and shortest paths

For modeling decision-making, we consider deterministic finite Markov decision processes (MDPs) [26] of the form $(\mathcal{S}, \mathcal{A}, \mathcal{T}, \mathcal{C})$, where $\mathcal{S} = \{s_1, \dots, s_N\}$ and $\mathcal{A} = \{a_0, \dots, a_M\}$ are the finite state and action space, respectively. $\mathcal{T} : \mathcal{S} \times \mathcal{A} \to \mathcal{S}$ denotes the state transition function and $\mathcal{C} : \mathcal{S} \times \mathcal{A} \to \mathbb{R}$ is the cost function. If all components of the MDP are known, standard offline reinforcement learning (RL) algorithms for MDP planning can be used to learn the optimal value function to find a policy $\Pi : \mathcal{S} \times \mathcal{A} \to [0, 1]$ minimizing the long-term cost.

Deterministic finite MDPs can be equivalently modeled as graphs $\mathcal{G} = (\mathcal{V}, \mathcal{E})$, where the nodes $\mathcal{V} = \mathcal{S}$ represent states and the edges $\mathcal{E} = \{(s, s') \mid \exists a \in \mathcal{A} : T(s, a) = s'\} \subseteq \mathcal{V} \times \mathcal{V}$ denote possible transition between them [45, 46]. An equivalent cost function $\mathcal{C} : \mathcal{E} \to \mathbb{R}$ specifies the cost for each transition. The optimal policy for navigating between two states can be obtained by solving the shortest path problem. This problem consists of finding the shortest path between the two nodes such that the sum of the visited edge costs is minimized [47]. By applying the dynamic programming principle, the single-source shortest path problem for non-negative edge costs can be efficiently solved using Dijkstra's algorithm [48]. The A* algorithm uses a heuristic to guide the search to increase efficiency. D* lite [49] is a dynamic version of A*, which can efficiently update the shortest path solution if costs of the graph change.

### 2.2 Bayes-adaptive MDPs

In Bayes-adaptive MDPs (BAMDPs), the goal is to solve an MDP while being uncertain about its transition kernel $\mathcal{T}$ [50]. During interaction, the belief $\phi \in \Phi$ about the unknown transition function parameters $\theta \in \Theta$ is updated using the observed information $\mathcal{I}$ about the transition via Bayes rule, $P(\theta \mid \mathcal{I}) \propto P(\mathcal{I} \mid \theta)P(\theta)$. A BAMDP can be formally defined via state augmentation as an MDP $(\mathcal{S}', \mathcal{A}, \mathcal{T}', \mathcal{C}')$, where $\mathcal{S}' = \mathcal{S} \times \Phi$ is the set of hyperstates, capturing jointly the state as well as the belief about $\theta$. The augmented transition function $\mathcal{T}' : \mathcal{S}' \times \mathcal{A} \to \mathcal{S}'$ specifies how the state and belief jointly evolve. The quantities $\mathcal{A}$ and $\mathcal{C}'$ coincide with the underlying MDP. In principle, the augmented MDP specifies the optimal policy $\Pi$, but its computation is intractable except for a few special cases [51, 52]. Approximate methods have been proposed [53], such as BEETLE [54], which approximates the value function of the augmented MDP but remains tractable only for low-dimensional problems (in the original work, MDPs with up to nine states). Commonly, scalable approximations either modify the cost function to visit states that are unknown (exploration bonus) [55–57], or act optimistically in the face of uncertainty, e.g., by applying Thompson sampling [58, 59], or combinations of both [60, 61]. For shortest path problems in the robotic domain, it was suggested to act purely optimistically [49].

# 3 Method and model

We propose an approximate probabilistic inference method to estimate an agent's latent knowledge about transitions in the world based on a single observed navigation trajectory. Specifically, we model the agent's knowledge in a binary form using a vector $\tilde{\mathbf{k}} = (\tilde{k}_1, \ldots, \tilde{k}_N)$, where each $\tilde{k}_i \in \{0, 1\}$ indicates whether state $s_i$ in the MDP is known ($\tilde{k}_i = 1$) or unknown ($\tilde{k}_i = 0$) to the agent. If a state $s_i$ is known, we assume the agent has access to its transitions, and denote the set of all these known transitions by $\mathcal{E}^{\mathrm{k}} = \{(s_i, s_j) \in \mathcal{E} \mid \tilde{k}_i \vee \tilde{k}_j\} \subseteq \mathcal{E}$. For unknown states, the agent instead considers a set of potential transitions $\mathcal{E}^{\mathrm{pot}} \subseteq \mathcal{V} \times \mathcal{V}$, with potential costs $\mathcal{C}^{\mathrm{pot}} : \mathcal{E}^{\mathrm{pot}} \to \mathbb{R}$, which are believed to exist with probability $q$. The agent's knowledge evolves over time: it begins with the initial knowledge vector $\tilde{\mathbf{k}}$, and states become known as soon as they have been visited. This knowledge formulation can be easily extended, for example, to model agents that acquire knowledge of nearby states within a certain radius, or existence probabilities depending on the structure of the environment. In the following, we formalize the knowledge inference problem, i.e., to reason about a latent $\tilde{\mathbf{k}}$, and describe how to compute a sample-based solution given a likelihood and prior model with a single evaluation per sample. Then, we show how to incrementally compute the likelihood and the prior by leveraging an approximate solution to the agent's planning problem. An implementation of our algorithm is publicly available under the MIT Licence[1]. For the planning algorithm, we adapted an implementation[2] of D* lite, also available under the MIT License.

## 3.1 Bayesian knowledge inference

For reasoning about the agent's latent knowledge $\tilde{\mathbf{k}} = (\tilde{k}_1, \ldots, \tilde{k}_N)$ about the world, we model our belief about $\tilde{k}_i$ using a corresponding Bernoulli random variable $K_i \sim \mathrm{Bernoulli}(p_i)$, and define $\mathbf{K} = (K_1, \ldots, K_N)$. Given an observed trajectory $\mathbf{D} = (\tilde{s}^1, \ldots, \tilde{s}^T)$, our goal is to infer the posterior distribution $P(\mathbf{K} \mid \mathbf{D})$. Bayes' theorem gives $P(\mathbf{K} \mid \mathbf{D}) \propto P(\mathbf{D} \mid \mathbf{K}) P(\mathbf{K})$, showing that with a likelihood and prior model, the posterior can be determined. Exact inference is intractable, as computing the normalization factor requires summing over all $2^N$ possible values of $\mathbf{K}$, which scales exponentially in the number of states $N$.

To approximate the posterior, we employ Gibbs sampling [62]: At each iteration, we sample a single variable $K_i$ from $P(K_i \mid \mathbf{K}_{-i}, \mathbf{D})$, where $\mathbf{K}_{-i}$ denotes the vector of all knowledge variables except $K_i$, and $\mathbf{K}_{-i}$ takes values of the previous sample $\mathbf{k}^{\mathrm{prev}}$. For Bernoulli $K_i$, each new sample $\mathbf{k} = (k_1, \ldots, k_N)$ either remains equal to the previous sample $\mathbf{k}^{\mathrm{prev}}$ or differs by a single bit flip at index $i$. The probability for a bit flip is determined by the full conditional distribution

$$P(K_i = \neg k_i^{\mathrm{prev}} \mid \mathbf{K}_{-i}, \mathbf{D}) = \left(1 + \frac{P(\mathbf{D} \mid K_i = k_i^{\mathrm{prev}}, \mathbf{K}_{-i})}{P(\mathbf{D} \mid K_i = \neg k_i^{\mathrm{prev}}, \mathbf{K}_{-i})} \frac{P(K_i = k_i^{\mathrm{prev}}, \mathbf{K}_{-i})}{P(K_i = \neg k_i^{\mathrm{prev}}, \mathbf{K}_{-i})}\right)^{-1}, \quad (1)$$

see Section A.1 for the derivation. This formulation is computationally efficient: To produce a sample, we only need to evaluate the prior ratio and a single likelihood $P(\mathbf{D} \mid K_i = \neg k_i^{\mathrm{prev}}, \mathbf{K}_{-i})$, since the likelihood of the past sample, $P(\mathbf{D} \mid K_i = k_i^{\mathrm{prev}}, \mathbf{K}_{-i})$, was already computed in the previous iteration. However, even with this formulation, the inference procedure remains challenging: Each likelihood evaluation involves solving a planning problem under partial knowledge, corresponding to a BAMDP, which is commonly intractable. In Section 3.2, we present a tractable approximation of the likelihood, which can be efficiently computed for the derived Gibbs update in an incremental way. In Section 3.3, we describe a prior model suitable to navigation tasks. The complete algorithm for the knowledge inference procedure is given in Algorithm 1.

## 3.2 Planning model and likelihood

The likelihood of a trajectory given the agent's initial knowledge is given by $P(\mathbf{D} = (\tilde{s}^1, \ldots, \tilde{s}^T) \mid \mathbf{K}) = \prod_{t=1}^{T-1} \pi_t(\tilde{s}^t, \tilde{s}^{t+1})$, where $\pi_t : \mathcal{S} \times \mathcal{S} \to [0, 1]$ denotes the agent's policy representing the probability to transition from state $\tilde{s}^t$ to $\tilde{s}^{t+1}$ at time step $t$. For fully informed agents in deterministic finite MDPs, the optimal policy is to follow deterministically the shortest path to the goal [45]. To model stochastic decision-making of planning humans, prior work has modeled

---

[1]`https://git.rwth-aachen.de/bcs/projects/msch/bayesian-knowledge-inference`
[2]`https://github.com/Sollimann/Dstar-lite-pathplanner`

---

**Algorithm 1** Bayesian knowledge inference

---

**Output:** Samples $\{\mathbf{k}^1, \ldots, \mathbf{k}^L\}$ from the distribution $P(\mathbf{K} \mid \mathbf{D})$
**Input:** Number of samples $L$, Partial world model $W$, Trajectory $\mathbf{D}$
 1: Initialize $\mathbf{k}^0$ with $N$ random boolean elements
 2: Compute likelihood of $\mathbf{k}^0$ using Algorithm 2
 3: **for** $i$ in $\{1, \ldots, L\}$ **do**
 4:    Select $j$ from $\{1, \ldots, N\}$ randomly
 5:    Initialize $\mathbf{k}^{\text{switch}} \leftarrow \mathbf{k}^{i-1}$
 6:    Switch the value $k_j^{\text{switch}}$
 7:    Compute likelihood for $\mathbf{k}^{\text{switch}}$ by updating the past shortest path result (Algorithm 2)
 8:    Compute prior fraction for $\mathbf{k}^{\text{switch}}$ using Eq. (3)
 9:    Compute acceptance probability $p$ using Eq. (1)
10:    With probability $p$, set $\mathbf{k}^i \leftarrow \mathbf{k}^{\text{switch}}$, otherwise $\mathbf{k}^i \leftarrow \mathbf{k}^{i-1}$
11: **end for**

---

agents using softmax policies [42, 38, 8, 39]. In particular, Chandra et al. [39] model the probability of moving from state $s$ to a $s'$ if there is a corresponding edge as

$$\pi(s, s') \propto \exp\left(\beta V_s(s')\right), \quad \text{with} \quad V_s(s') = C(s, g) - C(s', g), \tag{2}$$

where $C(s, g)$ denotes the cost of the shortest path from $s$ to goal $g$, which is computed using the A* algorithm [63], and $V_s(s')$ describes the reduction of in shortest path length towards $g$. To extend this model to our problem setting, we face two key challenges: First, the agent has only partial knowledge of the environment, turning the problem into a BAMDP. Exact planning in this setting is in most cases infeasible, as the number of belief states grows exponentially with the number of unknown states. Second, the Gibbs sampling procedure from Section 3.1, requires evaluating a likelihood for each sample to draw. Since the agent's knowledge evolves while acting, the policy must be recomputed at every time step with the updated knowledge, requiring a large number of policy computations.

We address the first problem by extending the stochastic policy model from Eq. (8) to approximate planning under partial knowledge: Instead of computing the shortest path under full knowledge, we model the agent to plan based on expected costs under probabilistic transitions reflecting their uncertainty. Specifically, we assume that uncertain transitions succeed with probability $q$, leading to an expected cost of $\mathcal{C}^{\text{pot}}(s, s')/q$ for a transition from $s$ to $s'$, while known transitions retain their true cost $\mathcal{C}(s, s')$. This approximation can be interpreted as planning optimistically with a penalization for uncertainty that is proportional to the risk of failure. Importantly, it assumes that planning is based on the current belief only, ignoring effects of knowledge acquisition along the path. We use these expected costs for the softmax policy in Eq. (8) and assume that the agent replans at each time step based on its updated knowledge.

To mitigate the second challenge, the excessive recomputation of policies, we observe that the knowledge vector $\mathbf{k}$ is updated only at one position both between sampling iterations and trajectory steps. This leads to a modification of only a relatively small number of graph edges in the graph. We take advantage of this by employing a dynamic shortest path algorithm, specifically the D* lite algorithm [49] instead of A*. This enables the reuse of past intermediate results and reduces the computational cost for policy computations significantly. Section A.2 provides a more detailed derivation of the likelihood and policy model, and presents the full algorithm for incrementally computing the likelihood (Algorithm 2).

### 3.3 Spatial prior

For the prior model, we make the plausible assumption that a subject is more likely to possess information about a state if nearby states are known, for example because the subject has visited this region. To capture this spatial correlation of knowledge values, we use the Ising model [64]. The conditional probabilities for the Ising model are given by

$$P(K_i = k_i \mid \mathbf{k}_{-i}) = \prod_{s_j \in \text{Adj}_{\text{pot}}(s_i)} \exp(\psi_{ji}(k_i \mid k_j)), \quad \text{with} \quad \psi_{ji}(k_i \mid k_j) = 2\beta \sum_{k_j} (-1)^{k_i} \cdot (-1)^{k_j},$$

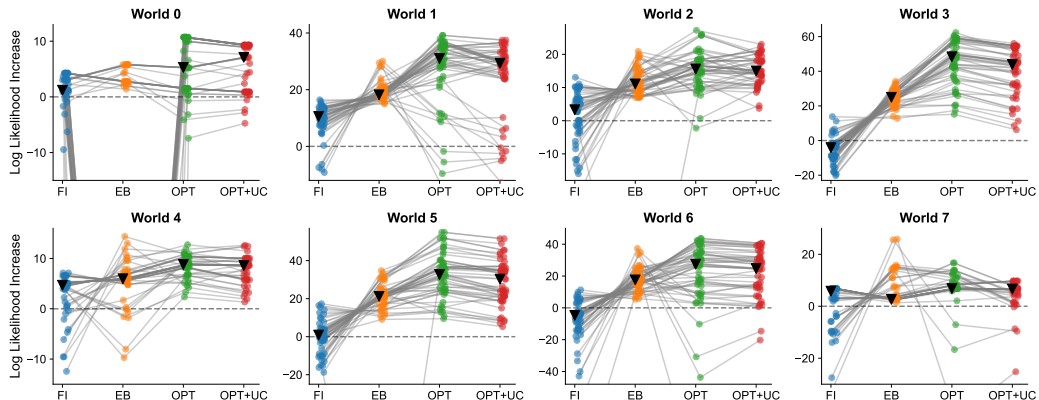

Figure 1: **Evaluation of the planning models on human data.** The graphs show the difference of log likelihood of all transitions between the random actor model and each respective planning model for individual trajectory (depicted as dots). Positive values indicate an improvement compared to the random actor model. The black triangles represent medians. For most trajectories, the optimistic models demonstrate improvements over both the random actor and uncertainty bonus model. For an extended figure of World 0, see Fig. 11 in the appendix.

where we defined $\mathbf{k}_{-i} = \mathbf{k} \backslash \{k_i\}$. As $\psi_{si}(\neg k_i, k_s) = -\psi_{si}(k_i, k_s)$, we obtain for the prior fraction in Eq. (1)

$$\frac{P(K_i = k_i^{\text{prev}} \mid \mathbf{K}_{-i})}{P(K_i = \neg k_i^{\text{prev}} \mid \mathbf{K}_{-i})} = \exp(-2\psi_{si}(k_i \mid k_s)), \qquad (3)$$

which enables us to sample from the posterior distribution as outlined in Algorithm 1. This prior allows the incorporation of the structure of the agent's knowledge without significantly increasing computational cost, as its computation cost is negligible compared to planning during inference.

## 4 Evaluation

We evaluate our method on two domains: a series of grid-world navigation tasks and a parking problem. In the evaluation, we compare our proposed approximate model for planning under partial knowledge against several other planning models: The optimal policy for a fully informed agent (FI), a random actor model (R), a model based on exploration bonus (EB), a purely optimistic model (OPT), and our approximate model (OUR) from Section 3.2, which can be interpreted as an optimistic model with penalization for uncertainty. Detailed information about these planning models is provided in Section A.7, and hyperparameters are listed in Section A.8. An additional complexity comparison between our method and exact inference is provided in Section A.3.

### 4.1 Application to grid world navigation tasks

For the grid world navigation tasks, we designed 8 grid worlds, where only a subset of fields was sampled as initially known. There, the goal is to control an avatar from a start field to a target field using a minimum number of steps. To validate our method on human behavioral data, we developed a browser game that allowed human subjects to navigate through these partially occluded grid worlds. We conducted an online experiment with 52 participants recruited via the Prolific platform. The experiment was approved by the local institutional review board (IRB). Detailed information about the grid worlds and the experiment design is provided in Section A.4.

**Planning model**  To evaluate how accurately our proposed approximate planning model captures human behavior in comparison to other methods, we computed the likelihoods of the trajectories collected in the human experiment. Figure 1 shows the log-likelihood differences between the random actor model and each of the other planning models. The results indicate that, for most trajectories, our proposed approximation and the optimistic planning model achieve higher log-likelihoods than the



Figure 2: **Models for planning under uncertainty.** Subjects need to navigate through the partially visible maze from a start field (smiley) to a target (flag) using a minimum number of steps. Hatched fields indicate initially visible fields. (FI) The rational agent with complete world knowledge follows the shortest path on the left side. (R) The random agent shows undirected behavior and only reaches the goal after a large number of steps. (EB) The model with exploration bonus explores locally until it discovers the path to the goal. (OPT) A purely optimistic agent plans as if all unknown fields were navigable and heads towards the goal if uncertain. (OUR) Our proposed approximation, which is optimistic with a penalization for uncertainty, initially navigates towards the goal due to its uncertainty but prefers the known path thereafter. This trajectory was most frequently observed in our online experiment.

other models. A Wilcoxon signed-rank test confirms that these differences are statistically significant for nearly all comparisons, with the exception of the exploration-bonus model in World 0 and the purely optimistic model in World 7 (see Section A.6.1 for details). Although this analysis provides evidence for differences between models, it does not explain why penalizing uncertainty is important for modeling human behavior in these grid worlds.

To qualitatively differentiate the planning strategies, we designed one of the eight grid worlds (World 0) to be smaller ($6 \times 5$) and structured such that each planning model predicts distinct behaviors. Fig. 2 shows the grid world along with the predicted trajectory when the most likely action is taken at each decision. Matching these unique trajectories to the online experiment data gave the following results: 21% of the subjects chose path FI, 19% chose path EB, 21% chose path OPT, and most participants, 35%, chose path OUR, representing our proposed approximate planning model. 4% of the trajectories did not fit any of the trajectories. Note that subjects who followed path FI could not have done so based on complete knowledge; their choice may have been influenced by variability or the belief that the "most obvious" path would be blocked, based on their previous trial experience.

**Inference model** To evaluate our proposed inference approach, we empirically compared it against two baselines. The first one is a simple agnostic model that assumes all fields are known with equal probability of 0.5, except for the start and goal positions. The second baseline additionally assigns a probability of 0.1 to a field where the agent turns around, based on the reasonable assumption that at these points the agent received new information. While our approach generates samples showing correlations between the fields, for the evaluation, we only consider the empirically estimated marginal probabilities. Consequently, our evaluation likely underestimates the information contained in the samples about the true posterior. We evaluate our method on two datasets: (1) trajectories from simulated agents using our proposed approximate planning model, and (2) trajectories collected in our online experiment. Since we controlled which fields were initially visible to participants, we can compare the inferred beliefs against the ground-truth knowledge.

We applied the inference approaches to each trajectory to estimate the probabilities of the fields being known. Figure 3 presents the results for two grid worlds (0 and 4). Although there are occasional false beliefs due to the high variability of the subjects' behavior, the inferred explanations produced by our method appear reasonable. Additional inference results for the other worlds and more human trajectories can be found in Fig. 15 in Section A.6. For quantitative evaluation, we assessed the likelihood of the true knowledge variables under the inferred marginal knowledge beliefs, shown in Fig. 4. On simulated data, our method shows clear improvements over both baselines, and a Wilcoxon signed-rank test confirms that these differences are significant (see Section A.6.1 for details). On experimental data, the quantitative results are more mixed: while we observe overall improvements in most environments, there are exceptions due to the high variability in participants' trajectories. The Wilcoxon signed-rank test indicates significant differences in five of the eight worlds. Inference for

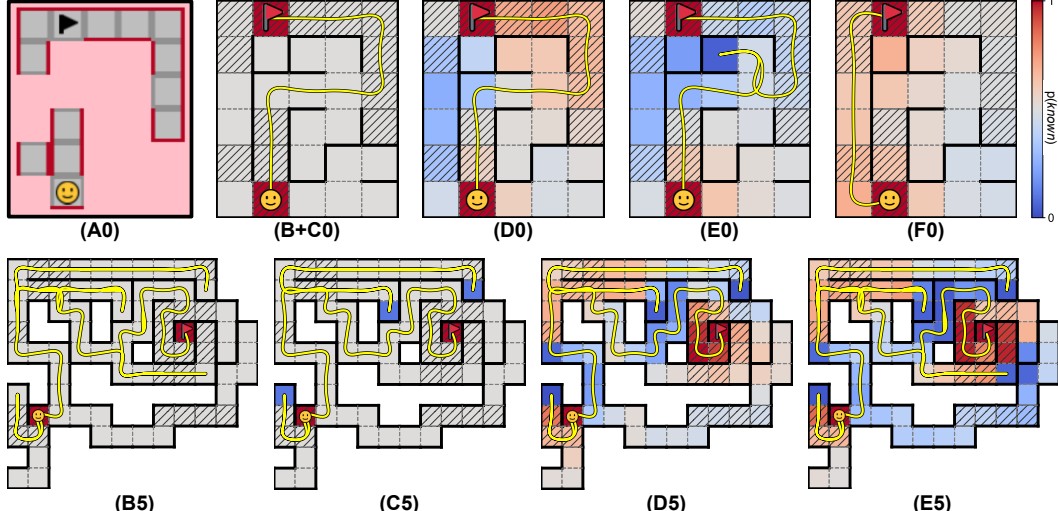

Figure 3: **Inferred knowledge beliefs of the considered inference methods.** (A) Subject's initial view of World 0. (B-E) Inferred beliefs (probability that a field is known) for grid worlds 0 (top) and 5 (bottom). Panels (B, C, D) display trajectories when following the most probable actions of the optimistic policy with uncertainty cost. Panel (B) shows the agnostic baseline, (C) the baseline based on turnarounds, and (D) our proposed method. In (D0), it infers that the fields on the left side must have been partly unknown to the subject, as well as the two fields beneath the goal field, otherwise the agent would have probably taken the path on the left. Panel (E) shows randomly sampled trajectories from the probabilistic policy, where in (E0), the inference method is certain that the subject could not have known the field diagonally to the goal. (F0) shows the trajectory of subject "hgryw", where the probability of the left side and the fields beneath the goal incorrectly receive a high probability. In (E5) the agent makes an additional detour due to the randomness in the policy, revealing that it did not know the top path to the goal and the dead end.

World 7 was particularly challenging, as the path to the goal was straightforward, so that knowledge had a very limited effect on behavior and behavioral variability leads to false beliefs.

To evaluate the efficiency gains resulting from reusing past results using the dynamic graph algorithm, we ran inference on the trajectories of 50 simulated agents in world 1, using the A* algorithm[3] instead of D* lite. Generating 1,000 samples on world 0 took an average of 7.48 hours (SD 1.53 hours) with A*, compared to 42 minutes (SD 5 minutes) with D* lite, representing an average time reduction factor of 10.7 times. Details about the used hardware used for these experiments are provided in Section A.8.

## 4.2 Application to the parking problem

While exact planning in BAMDPs is usually intractable, the parking problem [65] is an exception where the task structure allows for an exact solution. We use this task to evaluate our inference method with approximate planning against the true optimal policy. In this task, a driver proceeds along a sequence of parking spots, each potentially free or occupied, with parking costs decreasing along the route. However, if the driver passes the last parking spot without having parked, they need to park in the garage at a high cost. Importantly, the availability of each parking spot is unknown until the driver reaches it, making this problem a BAMDP. A formal description of the task and the exact solution are provided in Section A.5.

We first evaluate how well the considered planning models approximate the exact policy. For randomly sampled parking occupancies, we compute the policies for all models at each parking spot (Fig. 5A) and report their mean KL divergence from the exact optimal policy (Fig. 5B). Our proposed model closely approximates the true optimal policy by reaching the lowest KL, with the exploration bonus model as the next-best approximation. Next, we assess how the choice of the planning model

---

[3] https://pypi.org/project/astar/

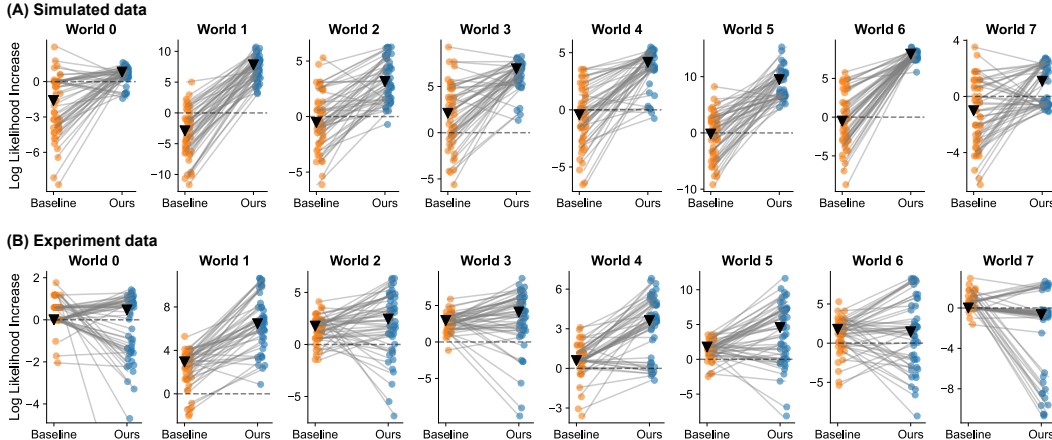

Figure 4: **Evaluation of the knowledge inference approach.** The graphs show the difference in log likelihood of the true knowledge information between an agnostic model and the inference methods for individual trajectories (depicted as dots) based on the marginal belief for each field. Positive values indicate an improvement compared to the agnostic model. The black triangles represent medians. For simulated data, our method clearly shows improvements over both the agnostic model and the heuristic. Despite the high variability in the experimental data causing numerous false beliefs, our method demonstrates overall improvements over the baseline. World 7 was particularly challenging for the inference as the optimal path was straightforward and therefore uninformative.

affects inference performance. Since all policies in this task can be computed efficiently, we run the knowledge inference procedure using each planning model to perform inference on trajectories generated using the exact optimal policy. Inference results using our proposed approximate planning model are shown in Fig. 5C. Similar to the grid world setting, we also compare against agnostic beliefs (AGN) and a manually designed heuristic baseline (HEUR). This heuristic assigns free parking spots ahead of chosen parking a 0.1 probability of being known, while other spots are known with belief 0.5. Figure 5D shows the average KL divergences of the inferred knowledge beliefs between the exact policy and other planning models. We find that both our proposed planning model and the exploration bonus lead to inference accuracies very close to the exact planning model, since the exploration bonus trades off garage cost and potential rewards of late parking similarly in this task.

## 5   Conclusion and future work

In this paper, we have addressed the problem of inferring the knowledge of an observed agent that acts under partial knowledge about the structure of the environment. We formulated the problem as an approximate Bayesian inference task and proposed an approximate planning model that allows for efficient evaluation within the inference procedure. We evaluated the method on both artificially generated data and data obtained from an online experiment using two task domains. By exploiting the incremental structure of the agents' knowledge using D* lite, we achieved a significant speedup compared to naively using the standard A* planning algorithm. While our approximate planning model proved effective for modeling the exact policy, the benefit of uncertainty penalization in modeling real human behavioral data still requires further investigation. The evaluation results suggest that our inference approach can provide a reasonable belief about the agent's world knowledge, particularly where it strongly influences behavior. As a next step, our method could be applied to analyze, for example, how subjects' awareness depends on topological features when solving mazes under time pressure. It could also be readily used to study spatial memory by examining which parts of a larger maze participants recall based on its local structure.

Our approach comes with several limitations. Even if the approach is tailored to medium-sized state spaces, the approach is unlikely to scale to very high dimensions in its current form. Further scaling to larger state and belief spaces, and beyond binary knowledge representations, for example by employing amortized inference or advanced sampling methods, would be an interesting direction for future research. Currently, all unknown transitions are modeled with a single probability of

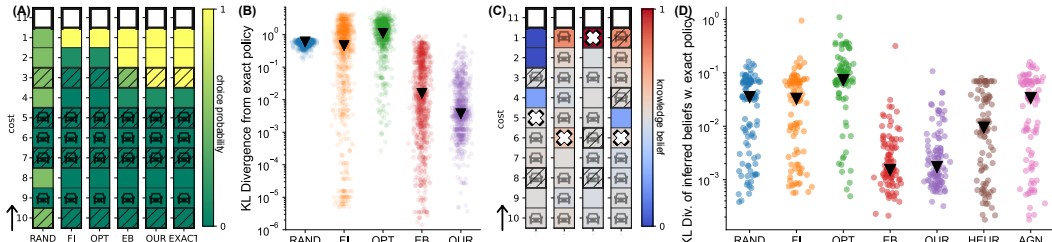

Figure 5: **Evaluation of the parking problem.** (A) Choice probabilities for the considered planning models. Hatched fields indicate initially known parking spots. (B) Mean KL divergences of planning models from the exact optimal policy for generated problems. The black triangles indicate medians. (C) Inferred knowledge beliefs based on simulated parking decisions using the exact planning model. Crosses indicate the chosen spots. (D) Mean KL divergences between inferred beliefs using the exact and other planning models for generated problems. Both inference with our proposed planning model and exploration bonus lead to similar beliefs as inference under the exact model.

traversability, which simplifies inference but could be extended in future work to allow different probabilities for different structures, such as corridors and junctions. Moreover, although the inference approach already captures the correlated structure of beliefs, this structure remains mostly latent in the data. Making this correlation structure accessible and enabling efficient recomputation when given additional information (conditioning) would be beneficial. Also, investigating how additional information sources, such as eye-tracking data [66], could be used to increase the reliability of the approach would be a valuable direction for future research. Furthermore, future work could aim to consider planning in non-spatial domains, for example by using the PDDL formalism [7].

A common challenge in behavioral experiments is that participants' decisions often exhibit dependencies across trials and depend on prior experience. Furthermore, human subjects have been shown to learn about the structure of the experiment design [33, 34]. These factors could have introduced additional noise and biases to the experimental data. Improving experimental design to reduce these influences or explicitly modeling such influences hierarchically would increase the interpretability of the results. Future work could also extend our approach to more naturalistic settings, including stochastic environments, and explore incremental approaches for other types of planning [67]. Exploring different forms of the experiment, such as augmented reality, might also prove helpful in this regard.

## Acknowledgments and Disclosure of Funding

We thank Dominik Straub, Fabian Tatai, and Philipp Fröhlich for insightful discussions. This work was supported by the Hessian Ministry of Higher Education, Research, Science and the Arts and its LOEWE research priority program 'WhiteBox' under grant LOEWE/2/13/519/03/06.001(0010)/77. Computations for the evaluation were performed on the high-performance computer Lichtenberg at the NHR Centers NHR4CES at Technische Universität Darmstadt.

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

# A Appendix

## A.1 Derivation of the switch acceptance probability

$$P(K_i = \neg k_i^{\text{prev}} \mid \mathbf{K}_{-i}, \mathbf{D}) = \frac{P(K_i = \neg k_i^{\text{prev}}, \mathbf{K}_{-i}, \mathbf{D})}{P(\neg k_i^{\text{prev}}, \mathbf{K}_{-i}, \mathbf{D}) + P(k_i^{\text{prev}}, \mathbf{K}_{-i}, \mathbf{D})} \tag{4}$$

$$= \left(1 + \frac{P(K_i = k_i^{\text{prev}}, \mathbf{K}_{-i}, \mathbf{D})}{P(K_i = \neg k_i^{\text{prev}}, \mathbf{K}_{-i}, \mathbf{D})}\right)^{-1} \tag{5}$$

$$= \left(1 + \frac{P(\mathbf{D} \mid K_i = k_i^{\text{prev}}, \mathbf{K}_{-i})}{P(\mathbf{D} \mid K_i = \neg k_i^{\text{prev}}, \mathbf{K}_{-i})} \frac{P(K_i = k_i^{\text{prev}}, \mathbf{K}_{-i})}{P(K_i = \neg k_i^{\text{prev}}, \mathbf{K}_{-i})}\right)^{-1} \tag{6}$$

## A.2 Further information on likelihood and planning model

The likelihood of the visited states $\mathbf{D} = \{\tilde{s}^1, \ldots, \tilde{s}^T\}$ given the knowledge values $\mathbf{k}$ decomposes as

$$P(\mathbf{D} = \{\tilde{s}^1, \ldots, \tilde{s}^T\} \mid \mathbf{K} = \mathbf{k}) = \prod_{t=1}^{T-1} \pi_t(\tilde{s}^t, \tilde{s}^{t+1}), \tag{7}$$

where $\pi_t : \mathcal{S} \times \mathcal{S} \to [0, 1]$ denotes the stochastic policy at time $t$, taking the form

$$\pi(s, s') \propto \exp\left(\beta(\tilde{\mathcal{C}}_t(s, g) - \tilde{\mathcal{C}}_t(s', g))\right). \tag{8}$$

Here, $\tilde{\mathcal{C}}_t : \mathcal{S} \times \mathcal{S} \to \mathbb{R}$ is the function measuring the expected shortest path cost between two states $s_i$ and $s_j$ at time $t$, formally defined as

$$\tilde{\mathcal{C}}_t(s_i, s_j) = \begin{cases} \mathcal{C}(s_i, s_j) & \text{if } (k_i^t \vee k_j^t) \wedge (s_i, s_j) \in \mathcal{E}, \\ \mathcal{C}^{\text{pot}}(s_i, s_j)/q & \text{if } \neg(k_i^t \vee k_j^t) \wedge (s_i, s_j) \in \mathcal{E}_i^{\text{pot}}, \\ \infty & \text{otherwise,} \end{cases}$$

where $k_i^t$ is the knowledge value for state $s_i$ updated for time $t$, which is 1 if $k_i = 1$ (the field was initially known) or if $s_i \in \{\tilde{s}^1, \ldots, \tilde{s}^{t-1}\}$, so the state has been visited before.

A naive computation of the policy for each sample and time step is computationally expensive. We observe that within one trajectory between two time steps, the agent gains only knowledge of the currently visited field $s_i$. If the field is assumed to be initially known, i.e., $k_i = 1$, the policy does not change and we can directly reuse the planning results from the previous time step. Furthermore, within the Gibbs sampling process, the knowledge value is switched only for one field $s_j$. If the field was part of the trajectory up to the current time step $t$, i.e., $s_j \in \{\tilde{s}^1, \ldots, \tilde{s}^t\}$, the agent knows the field at time $t$ in both the previous Gibbs sample and the current one. In this case, we can directly reuse the planning result from the past iteration. Otherwise, in comparison to the past sample, only the edges potentially connected to $s_j$ may change: For all $s_m \in \mathcal{S}$ with $(s_j, s_m) \in \mathcal{E}^{\text{pot}}$, $C(s_j, s_m)$ potentially increases if there is no edge between $s_j$ and $s_m$, i.e., $(s_j, s_m) \notin \mathcal{E}$, and may decrease otherwise. By using a dynamic shortest path algorithm such as D* lite, we can update the planning results from the past sample, updating the few affected edges. For the first sample, where we do not have previous planning results, we can reuse the planning results from the respective past time step (for iterations $t > 1$) instead. With this, we only need to solve the full shortest path problem once, for the first sample for $t = 1$. The complete algorithm for computing the likelihood is provided in Algorithm 2.

## A.3 Complexity analysis

In the following, we compare the complexity of our proposed algorithm to exact inference as used in Bayesian inverse planning. For exact inference, we must consider all $2^N$ combinations of values of $K$, where $N$ is the number of states. For each configuration, computing the agent's policy and likelihood results in a complexity of $\mathcal{O}(2^N \cdot C_{\text{planning}})$. Planning in the belief space using value iteration has complexity $\mathcal{O}(|\mathcal{S}|^2 \cdot |\mathcal{A}|/\epsilon) = \mathcal{O}(2^{2N} \cdot 4/\epsilon)$, with $\epsilon$ the precision of value iteration, since the belief MDP has $2^N$ states. Additionally, planning has to be performed up to $N$ times for each configuration,

**Algorithm 2** Likelihood computation

---

**Output:** Likelihood $P(\mathbf{D} \mid \mathbf{k})$, Planning results $\mathcal{P}$ for all $1, \ldots, T-1$
**Input:** Partial world model $W$, Knowledge values $\mathbf{k}$, Trajectory $\mathbf{D} = \{\tilde{s}^1, \ldots, \tilde{s}^T\}$,
      **optional:** Planning results of previous iteration $\mathcal{P}^{\text{prev}}$, Knowledge switching value $k_j$

1: **for** $t$ in $\{1, \ldots, T-1\}$ **do**
2:     Get state index $i$ from trajectory state $\tilde{s}^t$
3:     **if** $t > 1$ **and** $\mathbf{k}[i]$ **then**
4:         $\mathcal{P}[t] \leftarrow \mathcal{P}[t-1]$
5:     **else if** $\mathcal{P}_{\text{prev}}$ **and** $j \in \{\tilde{s}^1, \ldots, \tilde{s}^t\}$ **then**
6:         $\mathcal{P}[t] \leftarrow \mathcal{P}_{\text{prev}}[t]$
7:     **else if** $\mathcal{P}_{\text{prev}}$ **then**
8:         Update planning results $\mathcal{P}_{\text{prev}}[t]$ with switch at $j$
9:     **else if** $t > 1$ **then**
10:       Update planning results $\mathcal{P}[t]$ with switch at $i$
11:     **else**
12:       Compute planning results $\mathcal{P}[t]$
13:     **end if**
14:     Compute policy $\pi_t$ using Eq. (8) with $\mathcal{P}[t]$
15:     Compute likelihood of $\mathbf{D}[t+1]$ using Eq. (7)
16: **end for**

---

as the agent is assumed to replan along the trajectory (and then all fields have been visited, including the goal). The results in a total complexity for inference of $\mathcal{O}(N \cdot 2^{3N}/\epsilon)$.

In contrast, our Gibbs-sampling-based approach scales linearly with the number of variables: we sample each $K_i$ conditioned on all others, leading to a complexity of $\mathcal{O}(L \cdot N \cdot C_{\text{planning}})$, where $L$ is the number of samples. Our planning approximation avoids the complete belief space and uses the shortest path algorithm D* lite, which has worst-case complexity similar to Dijkstra, $\mathcal{O}(|\mathcal{E}| + |\mathcal{V}| \log |\mathcal{V}|) = \mathcal{O}(4 \cdot N + N \log N) = \mathcal{O}(N \log N)$. Similar to Bayesian inverse planning, planning needs to be done $N$ times per sample. Therefore, the total cost is only $\mathcal{O}(L \cdot N \cdot N \cdot N \log N) = \mathcal{O}(L \cdot N^3 \cdot \log N)$.

### A.4 Navigation experiment design and procedure

Using the Prolific platform, we recruited 52 participants from the US and UK who were fluent in English. The experiment took between 6 and 10 minutes and each participant was paid £1.70 plus a performance bonus of up to £1. The bonus was based on the reward earned during a randomly chosen trial. The reward was set to $100 - N_{\text{steps}}$ pence. Participants' earnings were greater than the minimum wage of £11.44/h (UK) and a total of £165.08 was spent on participant compensation. To participate in our study, subjects had to read and accept our information and declaration on data protection policy, allowing us to process and publish the behavioral data in anonymized form.

After presenting the introductory screens (Fig. 6) to the participants, the actual experiment began. In each trial of the experiment, subjects were shown a grid world, where only a subset of the fields was visible. The remaining fields were hidden and therefore unknown to the participants. The starting field and goal position were always initially visible (Fig. 7). The avatar could be controlled with the keyboard in up to four directions (left, right, up, down) and could only move if there was no wall in the chosen direction. There was a minimum interval between two successive movements of 300 ms. As soon as the avatar entered an unknown field, it became visible along with the surrounding walls and remained visible for the rest of the trial (Fig. 8).

Participants were instructed to reach the goal within a minimum number of steps to maximize their reward and, consequently, their bonus payment. Before the actual experiment, participants completed 7 introductory mazes to familiarize themselves with the task at varying levels of complexity (Fig. 9). These maps were not considered for bonus payments or for evaluation. The actual experiment consisted of 8 mazes, which were presented in a random order. At the end of the experiment, participants were asked to indicate their sex and age in a short survey and could leave a comment (Fig. 10).

Grid worlds 1-7 were manually designed by placing the start and goal positions at arbitrary locations, drawing a path between them (of varying complexity, for example, the path of grid world 1 is very

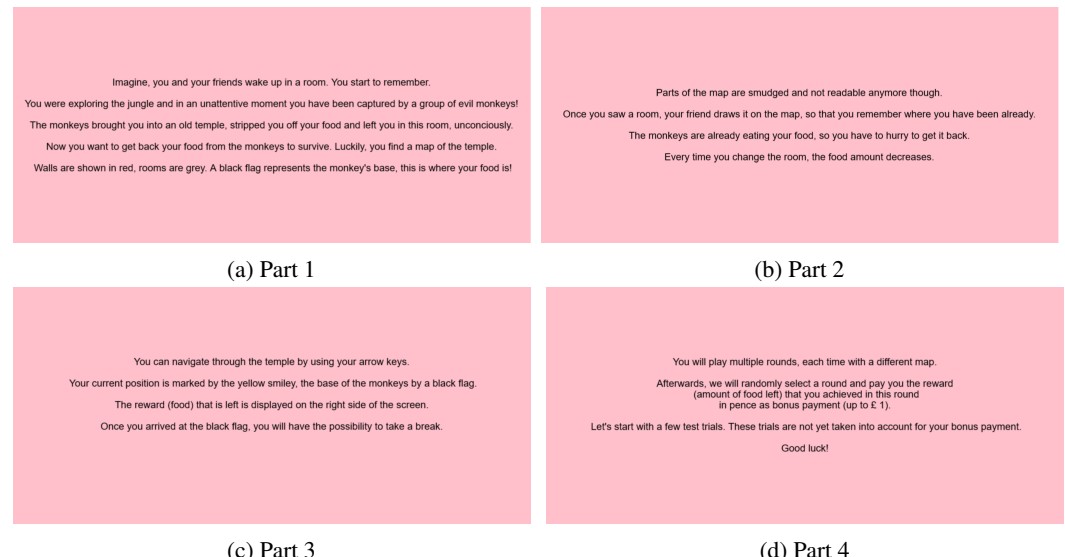

(a) Part 1            (b) Part 2

(c) Part 3            (d) Part 4

Figure 6: **Introduction displayed to the participants.** The introduction gives a cover story to explain the setup and goals. Participants were instructed how the avatar was controlled and details about the bonus payment were given. Participants could go back and forth in the introduction for the case they wanted to reread some passages.

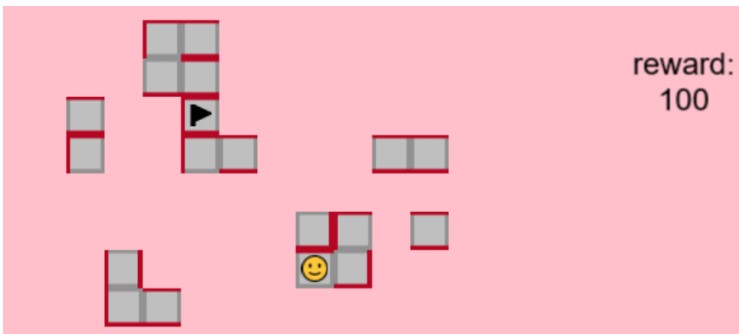

Figure 7: **World 6 before movement.** Only some fields of the world were visible right from the start, always including the start and goal field. Visible, walkable fields were grey, walls were red. The goal was marked by the black flag, the player's position by the yellow smiley. In this example, the player could walk to the right (known field), to the left, or down (unknown fields). Unknown fields were not visible upfront but became visible as soon as the player visited them. The reward, displayed on the right, was always 100 at the beginning of a level and reduced by 1 with each move.

complicated while for world 4 it is straightforward), and extending the path by dead-ends. We used all the worlds we designed in the experiment, providing transparency about the performance of our method by showing cases where beliefs are hard to identify (e.g., grid world 7 where the path had low complexity). Moreover, the varying complexity prevents humans from learning about the experiment's structure, e.g., learning to prefer indirect paths (which probably still occurred to some extent in the conducted experiment).

### A.5 Parking problem

The parking problem can be formally defined as a graph BAMDP $(\mathcal{V}', \mathcal{E}, \mathcal{C})$,

- hyperstates $\mathcal{V}' = \mathcal{V} \times \Phi$ consisting of states (nodes) $\mathcal{V}$ and knowledge states $\Phi$,
- states $\mathcal{V} = \{1, \ldots, M\} \cup \{\bot, G\}$, where $M$ is the number of parking spots, $\bot$ the terminal state and goal, and $G$ the garage state,

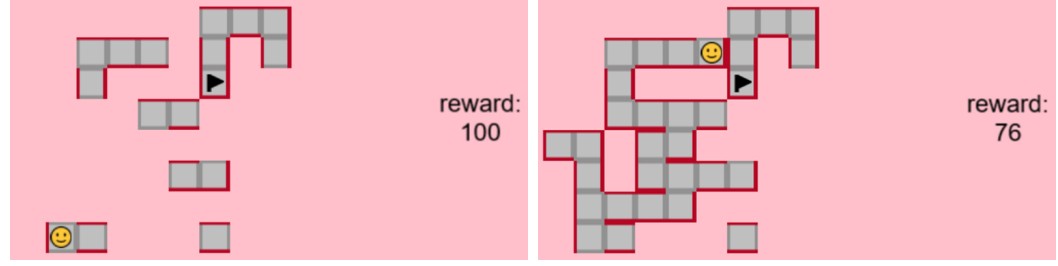

(a) View at the beginning        (b) View after navigating through parts of the world

Figure 8: **Example of visibility and reward changes upon movement.** At the beginning of the level, before moving, only some fields of the world were visible and the reward was set to 100. The start and goal fields were always visible right from the beginning. When moving through the world, unknown fields that have been explored remained visible throughout the whole trial. The reward is reduced by 1 with each move. The reward of 76 in (b) corresponds to 24 moves having been made.

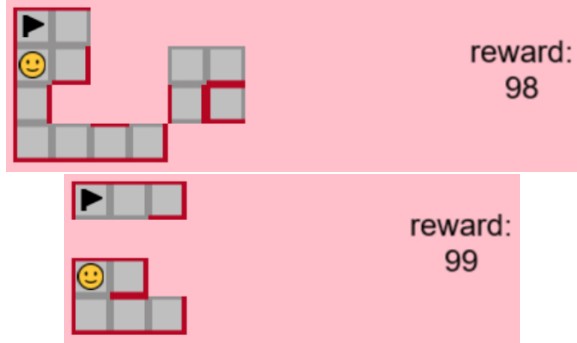

Figure 9: **Different world complexities.** Worlds could implement straight passages to the goal (*left*) or passages that consisted of multiple turns (*right*). The two shown worlds both had their starting points at the bottom left corner, the block above was not visible from the beginning. They were part of the test worlds with the purpose of familiarizing the participants with the concept of the experiment.

Please answer all questions carefully before clicking the button at the bottom to finish.

What is your age? -- select an option -- ⌄

What is your sex? -- select an option -- ⌄

Do you have anything you would like to note?

Finish survey

Figure 10: **Survey at the end of the experiment.** The survey consisted of questions about the participants and gave the possibility to leave additional comments. Selectable options for age were *<25*, *25-34*, *35-44*, *45-54* and *55+*. Selectable options for gender were *female*, *male*, and *other*. Both questions also had the option *prefer not to answer*.

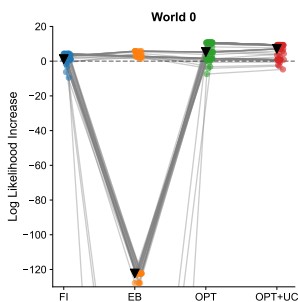

Figure 11: Evaluation of the planning models on human data for World 0 (extended plot)

- knowledge states $\Phi = \mathcal{P}(\{1, \ldots, M\})$, indicating the "known" parking spots, where the occupancy is known. Here, $\mathcal{P}$ denotes the power set operator.
- transitions $\mathcal{E} = \mathcal{E}_{\text{continue}} \cup \mathcal{E}_{\text{park}} \cup \mathcal{E}_{\text{garage}}$ , with
  - $\mathcal{E}_{\text{continue}} = \{(i, i+1), \text{ for } i = 1, \ldots, M-1\} \cup \{(M, G)\}$ the actions to continue to the next parking spots,
  - $\mathcal{E}_{\text{park}} = \{(i, \perp), \text{ for } i = 1, \ldots, M-1, \text{ if parking } i \text{ is free}\}$ the actions to park,
  - $\mathcal{E}_{\text{garage}} = \{(G, \perp)\}$ the action to park at the garage,
- the cost function $\mathcal{C}(s_i, s_j) = \begin{cases} 0 & \text{if } (s_i, s_j) \in \mathcal{E}_{\text{continue}}, \\ M - s_i + 1 & \text{if } (s_i, s_j) \in \mathcal{E}_{\text{park}}, \\ M + 1 & \text{if } (s_i, s_j) \in \mathcal{E}_{\text{garage}}, \\ \infty & \text{otherwise.} \end{cases}$

For the parking problem, all "continue" transitions are always known, while the "parking" transitions are only known if the respective parking spot $s_i$ is known, i.e., $k_i = 1$. In contrast to the formulation of Bertsekas [65], we assume that the agent has initial knowledge about some of the parking spots, which we aim to infer.

### A.6 Additional results

In the following we provide additional evaluation results.

#### A.6.1 Grid world tasks

For the grid world tasks, we present the following additional evaluation results:

- Additional plots showing the trajectories of the considered policies analogous to Fig. 2 for worlds 1-7 are shown in Fig. 14.
- An extended plot for the quantitative evaluation of the planning models (from Fig. 4) is shown in Fig. 11.
- Additional results of our proposed inference approach for additional worlds and subjects are provided in Fig. 15. Examples were picked to give an overview of the trajectory variabilities encountered in the experiment.
- To evaluate the influence of the prior, we conducted an ablation and performed inference using a uniform prior. The results were qualitatively very similar (see Fig. 12), but the relative improvements between the baseline and our method were on average half as large (average factor 1.991). This suggests that while the prior has some influence, it is not essential for the effectiveness of our method.

In addition, we performed statistical tests to investigate whether the differences observed in the evaluation were significant:

First, we considered the evaluation of the inference method. To choose a suitable test, we first checked the assumptions of normality for the datapoints of each world and method combination using the

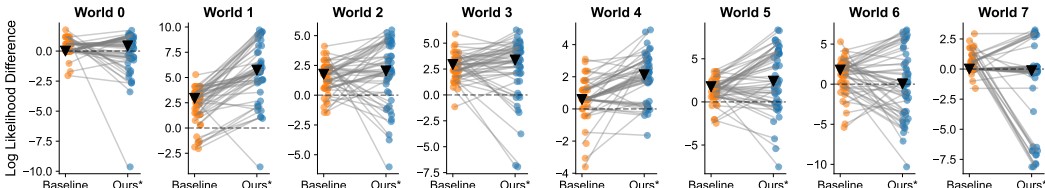

Figure 12: **Ablation study of inference on human navigation data with a uniform prior.** Ours$^*$ denotes our inference method with a uniform prior. In comparison to the results in Fig. 4, our method performs qualitatively very similarly, but the relative improvements between the baseline and our method were on average half as large (average factor 1.991).

Shapiro-Wilk test [68] with $p = 0.05$. We found that only a subset of the data (4/16) followed a normal distribution. Given that the t-test requires normality assumptions to be met, we applied the Wilcoxon signed-rank test [69]. This non-parametric test does not assume normality and can be used to assess the differences between the central values of the baseline and our method for each world. We applied a False Discovery Rate (FDR) correction using the Benjamini-Hochberg method [70] to control for multiple comparisons. Given that all worlds have individual belief spaces and represent different tasks, we consider them separately. The corrected p-values (q-values) of the Wilcoxon signed-rank test are shown in Table 1. For simulated data, for each world, significant differences between our method and the baseline were found. For the human experiment data, differences were significant for five of the eight worlds.

| world | simulated data | | real human data | |
|---|---|---|---|---|
| | q-value | significant | q-value | significant |
| 0 | 0.0000007368 | Yes | 0.09551332 | No |
| 1 | 0.0000000013 | Yes | 0.00000003 | Yes |
| 2 | 0.0000000172 | Yes | 0.05887162 | No |
| 3 | 0.0000000248 | Yes | 0.04011417 | Yes |
| 4 | 0.0000000016 | Yes | 0.00000008 | Yes |
| 5 | 0.0000000013 | Yes | 0.00214453 | Yes |
| 6 | 0.0000000013 | Yes | 0.62287947 | No |
| 7 | 0.0000110184 | Yes | 0.00325433 | Yes |

Table 1: Adjusted p-values and significance results on inference log likelihood differences between our approach and the baseline using the Wilcoxon signed-rank test.

For the planning models, we applied the same procedure. Again, the Shapiro-Wilk test only indicated normality for 19% of the planning model-world combinations. We therefore proceeded with the Wilcoxon signed-rank test to check for significant differences between our proposed model and other planning models. The resulting adjusted p-values (q-values) are shown in Table 2. The differences of all combinations were significant, except for the exploration bonus model in world 0, and the purely optimistic model in world 7.

### A.6.2 Parking problem

For the parking problem, we provide a figure comparing the log-likelihoods of the true knowledge values under the inferred beliefs relative to agnostic beliefs in Fig. 13. One can observe that inference using exact inference leads to similar likelihoods as when using the exploration bonus or our approximate method. We conclude that for the parking problem, these planning methods approximate the exact solution accurately.

### A.7 Planning model: Exploration bonus

For the exploration bonus model, we designed the following MDP: Transitions were only considered from known fields; from unknown fields all transition probabilities were set to zero. Entering a non-visible field led to an exploration reward of 1, reaching the goal to a reward of 10. The policy

| world | fully informed | | exploration bonus | | optimistic | |
|---|---|---|---|---|---|---|
| | q-value | significant | q-value | significant | q-value | significant |
| 0 | 0.0000127346 | Yes | 0.5892034940 | No | 0.0006017581 | Yes |
| 1 | 0.0000000290 | Yes | 0.0431046667 | Yes | 0.0006887471 | Yes |
| 2 | 0.0000000017 | Yes | 0.0003253850 | Yes | 0.5847222927 | No |
| 3 | 0.0000000017 | Yes | 0.0000000469 | Yes | 0.0000000028 | Yes |
| 4 | 0.0000000648 | Yes | 0.0000295632 | Yes | 0.0007460520 | Yes |
| 5 | 0.0000000023 | Yes | 0.0000076167 | Yes | 0.0000000529 | Yes |
| 6 | 0.0000000017 | Yes | 0.0092637150 | Yes | 0.0097015615 | Yes |
| 7 | 0.0185740168 | Yes | 0.2507408261 | No | 0.0018645117 | Yes |

Table 2: Adjusted p-values and significance results of planning log likelihood differences using the Wilcoxon signed-rank test.

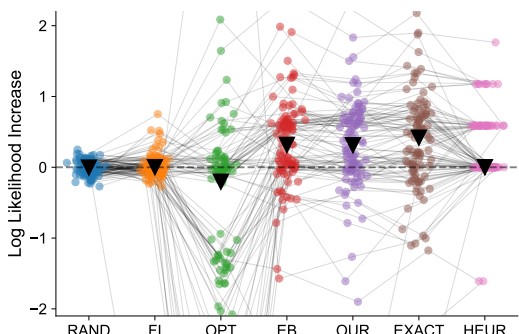

Figure 13: **Additional results for the parking problem.** Knowledge likelihoods under inferred beliefs when using different planning models for the inference method, relative to agnostic beliefs.

was formulated as a softmax over the Q function [26],

$$\pi(s, s') \propto \exp\left(\frac{1}{T}Q((s, s'))\right).$$

The Q function was computed using tabular value iteration [26]. For the value iteration, a maximum horizon of 100 steps was considered and it was previously stopped if the infimum norm of the Q function difference between two consecutive iterations was smaller than 0.01. The softmax temperature was set to 0.1.

For the parking problem, however, for each unoccupied state, the goal is directly accessible. With the previously described exploration bonus formulation, the agent would greedily follow transitions to the goal. Therefore, for this problem, we chose a different approximation: We plan similarly to our approximation, with the difference that instead of scaling the cost for uncertain decisions, agents incur a negative cost (reward) if they transition to an unknown field. This formulation makes agents "explore" unknown parking spots for a while, but park their car as soon as the high garage cost exceeds potential rewards through exploring new parking spots. This makes the exploration bonus strategy very similar to our approximation and the optimal policy for the parking problem, explaining the similar performance observed in the evaluation.

## A.8 Hyperparameters and hardware

Throughout the experiments, we used the following hyperparameters:

- Total number of samples for the inference: 1000
- Number of samples rejected for burnin phase: 100
- Parameter for the Ising model (spatial prior): $\beta = 0.2$
- Cost for transitions from/to known fields: 1

- Belief of traversability for unknown fields: $q = 0.5$
- Softmax temperature for our proposed planning model: 1
- Softmax temperature for the optimistic planning model: 0.8
- Computing resources used: Intel® Xeon® Platinum 9242 Processor (1 core per trajectory)

For the parking problem, we used the following hyperparameters:

- Number of parkings: 10
- Probability of a parking being free: 0.3
- Probability of knowing a parking spot: 0.3
- Belief of traversability for unknown fields: $q = 0.3$

The hyperparameters of sample sizes were selected such that the results did not vary anymore. Costs and temperatures were set to standard values, e.g., the cost for known transitions was set to 1 as in the work by Chandra et al. [39], and the cost for unknown transitions to twice as much, and the resulting trajectories were checked for plausibility. A low cost makes the agent more optimistic when acting under uncertainty (preferring unknown fields, potentially yielding shorter paths), while increasing the cost makes the agent more risk-averse, preferring known paths that might be longer. The hyperparameter of the prior was selected manually to produce moderately connected knowledge regions (neither fully fragmented nor overly clustered). The same hyperparameters were used both for generating the ground-truth knowledge values in the experiment (the fields initially displayed to the participants) and for inference. The temperature of OPT was set slightly smaller as for OUR, since the uncertainty cost leads to higher total costs, which would decrease stochasticity otherwise. For the grid world experiments, the parameter of the spatial prior was selected manually to produce moderately connected knowledge regions (neither fully fragmented nor overly clustered). The same hyperparameters were used both for generating the ground-truth knowledge values in the experiment (the fields initially displayed to the participants) and for inference.

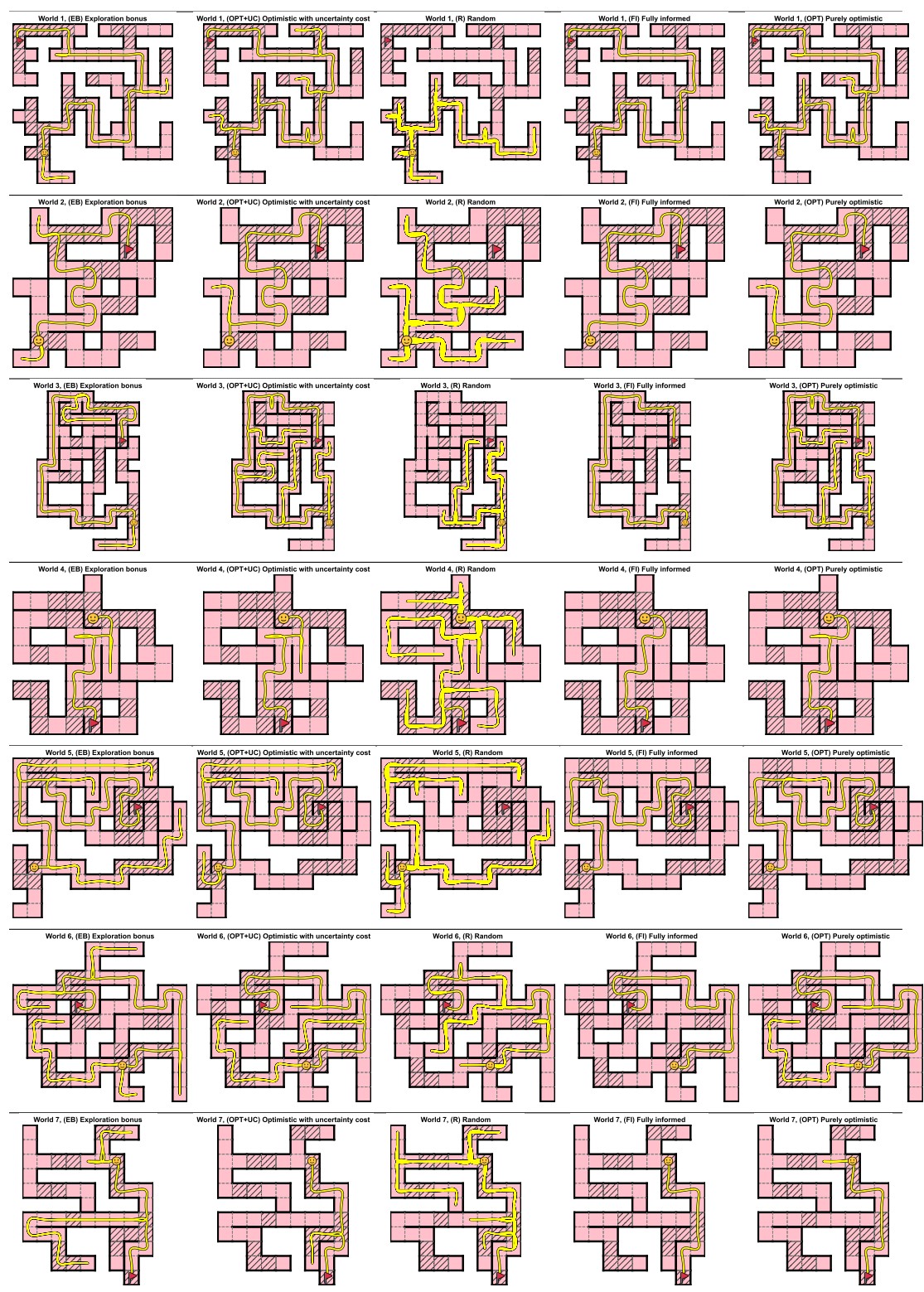

Figure 14: Models for planning under uncertainty for Worlds 1 - 7

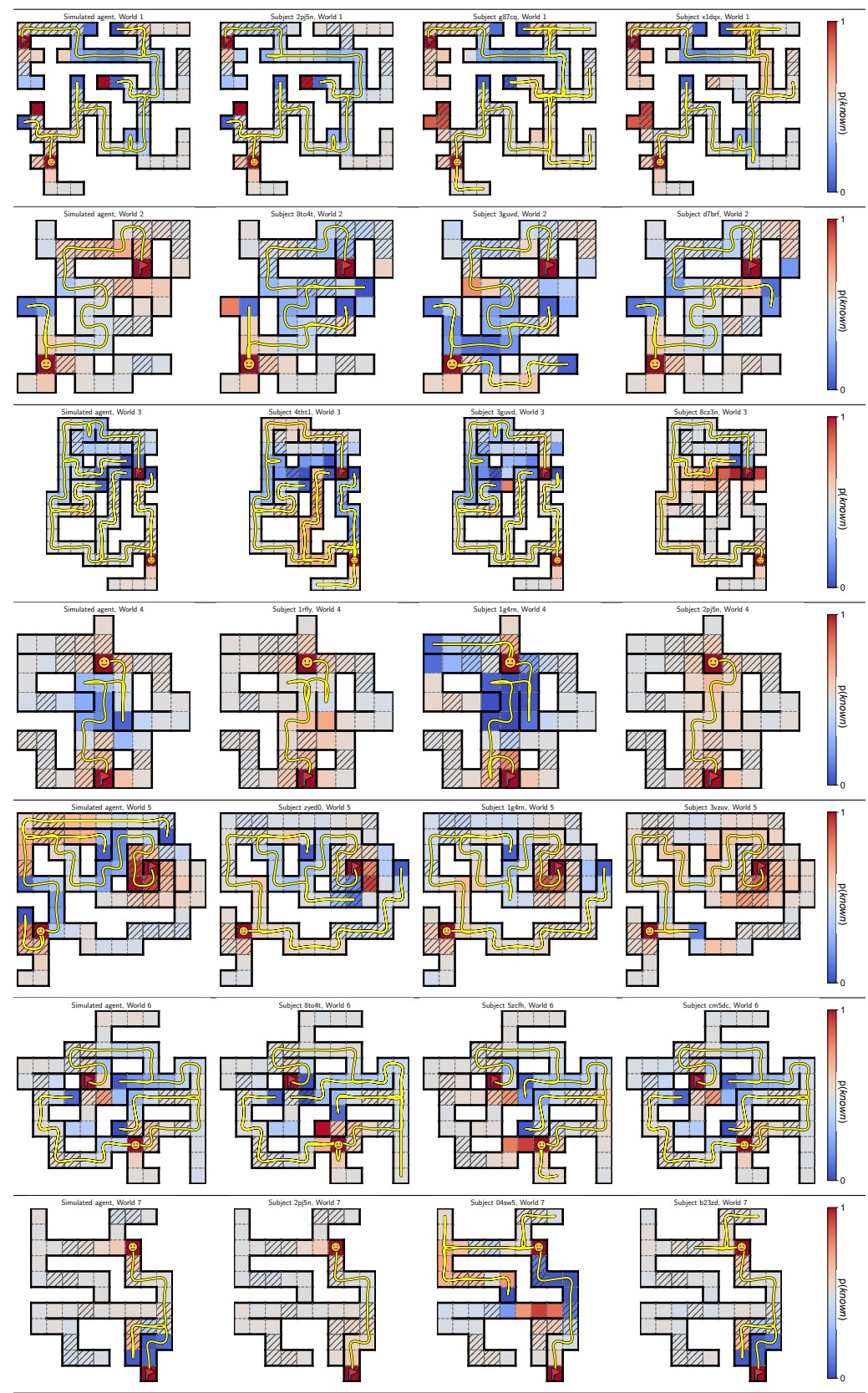

Figure 15: Additional results of our inference approach showing the variability of subjects' behavior

