# OpenReview forum: "What do you know? Bayesian knowledge inference for navigating agents"
_NeurIPS.cc/2025/Conference — NeurIPS 2025 poster_

### Official Review · Reviewer_cJ9W · 2025-06-06

**Clarity:** 4
**Significance:** 2
**Originality:** 3
**Rating:** 4
**Confidence:** 3

**Summary:**

This paper proposes a Bayesian approach for inferring the partial knowledge of agents in navigation scenarios. The approach posits a model where the transitions for a state are either known or unknown. Inference combines Gibbs sampling and D\* lite. Experiments in grid worlds and a parking scenario compare the proposed approach to a number of baselines with both simulated and real human data.

**Questions:**

1. Can you clarify how $q$  was set in experiments?
2. Can you further justify the spatial prior?

**Ethical Concerns:**

["NO or VERY MINOR ethics concerns only"]

**Final Justification:**

My only remaining concern is that the paper's impact may be limited by its narrow scope. But I think the paper's strengths outweigh its weaknesses.

**Limitations:**

yes

**Quality:**

3

**Strengths And Weaknesses:**

## Strengths

**Writing.** The paper is very clear and overall very well written. The organization is good and I did not find major typos or grammatical issues. It was also an enjoyable read.

**Systematic experiments**. The experiments are careful and systematic. There were several points while reviewing that I had a question that was then answered. Although the environments are relatively simple, they are similar to those used in related work.

**Novelty.** The paper does seem sufficiently novel, although the scope is a bit narrow (see below).

**Nice approach.** I like the idea of using D\* lite and I appreciate the empirical justification showing that it is much faster than using A\*.


## Weaknesses

**Scope.** While the paper does go beyond prior work by allowing for partial knowledge, it is also more narrowly scoped than prior work because of the focus on navigation and spatial tasks. For example, the approach of Zhi-Xuan et al. (2020), which I believe is the most similar prior work, also uses a combination of Bayesian inference and heuristic search, but they use a more general PDDL formalism that does not make any assumptions about navigation or space. The spatial assumption here seems fundamental to the model, which says that  the transition distribution for a state is either fully known or unknown---this makes sense for navigation, but maybe not so much for other environments with partial knowledge.

**Spatial prior.** I do not fully understand the need for the spatial prior as opposed to a uniform prior. Are empirical results worse without the spatial prior? Since the uniform prior is simpler than a spatial one, it would be good to run an ablation. It is also not ideal that the spatial prior introduces another hyperparameter, and it’s not clear how this hyperparameter was selected (I referred to the Appendix A.7 as well).

**Hierarchical Bayesian model.** The paper mentions the possibility that the human subjects may learn about the shared structure of the environments throughout the course of the experiment, but even within a single environment, it is conceivable that subjects are performing a kind of hierarchical inference where they are reasoning, for example, about the global probability of obstacles existing or parking slots being open. For example, looking at Figure 2, that kind of hierarchical inference could lead to the rightmost trajectory, as an alternative to the “optimistic with uncertainty penalization” model that is hypothesized in the paper.

**Minor: lines on unordered plots**. I don’t think the mean lines on the plots in Figures 4 and 5 are informative because there is no ordering relation between the approaches on the x axis. I can understand the motivation for including lines to connect individual participant data in Figure 4, but the mean lines don’t seem to make sense.

---

> ### Author Rebuttal · Authors · 2025-07-31
>
> We sincerely thank the reviewer for the thoughtful and constructive feedback. We are pleased that the clarity, novelty, and systematic experimentation of the paper were appreciated. Below, we respond to the reviewer’s specific comments and suggestions.
>
> ## Scope and spatial assumptions
> We agree that our formulation makes use of the spatial structure, which makes it particularly suited for navigation tasks. This structure is useful for two reasons: First, it provides a natural way to define optimistic planning over unknown transitions, enabling efficient inference. Second, it allows the use of heuristic search methods such as D* lite or A*, where spatial distance provides an effective heuristic.
> While the referenced work by Zhi-Xuan et al. (2020) uses a more general formalism based on PDDL, their approach also makes use of a spatial heuristic by using A* for planning and to limit search depth in their bounded rational planning model. Thus, both approaches implicitly exploit structured environments, though to different extents.
>
> While the spatial domain already offers a wide range of applications for our approach, we agree that extending our method to non-spatial domains could be an interesting and important future direction. We will note this in the revised future work section.
>
> ## Spatial prior
> The spatial prior captures the intuition that knowledge (or awareness/memory) of neighboring fields is likely to be correlated. For example, if a human knows one field in a corridor, they are likely to know neighboring fields as well. This prior allows the incorporation of the structure of the agent's knowledge without significantly increasing computational cost, as its computation cost is negligible compared to planning during inference.
>
> Importantly, our approach does not depend on the spatial prior. We have now conducted an ablation using a uniform prior and found that results were qualitatively very similar, but the relative improvements between the baseline and our method were on average half as large (average factor 1.991). This suggests that while the prior has some influence, it is not essential for the effectiveness of our method. We will include this ablation in the appendix of the final version. Nevertheless, in Bayesian inference, providing the flexibility to specify prior knowledge is conceptually important to ensure meaningful posterior beliefs.
>
> The hyperparameter of the prior was selected manually to produce moderately connected knowledge regions (neither fully fragmented nor overly clustered). The same hyperparameters were used both for generating the ground-truth knowledge values in the experiment (the fields initially displayed to the participants) and for inference. We will clarify this procedure in the appendix.
>
> ## Hierarchical Bayesian Inference
> We fully agree with the reviewer that human participants may perform higher-level reasoning about the structure of the tasks across and within environments, such as estimating the likelihoods of dead ends. As mentioned in our future work section, such hierarchical learning may account for the variability in our experimental data, including trajectories that appear irrational at first glance.
>
> Our work provides a foundational step to understand observed learning behavior by making inference about latent knowledge tractable for a relatively simple learning model. Hierarchical learning would considerably increase complexity, but could be a promising direction for future work. We will add this point to the future work section. This suggestion also aligns well with our broader research goals, which include investigating how subjects generalize knowledge across environments and whether parameters like $q$ (the belief over transition probability) change across maze topologies.
>
> ## Lines in Figure 4 and 5
> We appreciate the reviewer's comment that the black line connecting medians may be misleading, which was also noted by reviewer Sttp. We will remove it in the final version.
>
> ## Choice of hyperparameter $q$
> For the gridworld experiments, we set $q=0.5$ without tuning, to model an equal chance for success and failure for unknown transitions. In the parking experiment, we used $q=0.3$, which corresponds to the true probability of a parking spot being available (as used for generating the environments). This value was manually chosen to produce meaningful behavior when using the optimal policy: the agent should neither always park at the first nor last opportunity. We will clarify this setting in the final version.

---

> > ### Comment · Reviewer_cJ9W · 2025-08-02
> >
> > Thank you to the authors for their response. Overall, I remain positive about the paper. During this discussion period, I want to further clarify two points.
> >
> > On "Scope and spatial assumptions":
> > > While the referenced work by Zhi-Xuan et al. (2020) uses a more general formalism based on PDDL, their approach also makes use of a spatial heuristic by using A* for planning and to limit search depth in their bounded rational planning model.
> >
> > To the best of my knowledge, that work does not use a spatial heuristic. Note that A* does not necessarily refer to path planning, and in the case of PDDL, usually refers to planning with a heuristic derived from the PDDL model itself, e.g., through delete-relaxation methods as in the hFF heuristic among others.
> >
> > On "Spatial prior":
> > > We have now conducted an ablation using a uniform prior and found that results were qualitatively very similar, but the relative improvements between the baseline and our method were on average half as large (average factor 1.991).
> >
> > Thank you for the additional analysis. While the spatial prior makes intuitive sense in general, I don't think I understand why it helps in the specific experimental setup here. Is there something about distribution of the grid environments that makes the spatial prior help?

---

> > > ### Author Response · Authors · 2025-08-03
> > >
> > > __On "Scope and spatial assumptions":__
> > >
> > > Thank you very much for the clarification. We are not very familiar with PDDL-based planning frameworks, so your input is very valuable. Our earlier phrasing was imprecise, and we appreciate the correction. What we tried to express is that A* relies on a structured search space (such as a graph) and a heuristic function that estimates the cost to the goal. This heuristic does not need to be spatial, it could also be based on delete-relaxation, as in the hFF heuristic. In this sense, the planning process can be seen as "navigating" a structured space, even when it is not spatial in nature. This is also what we meant by saying that the approach also "exploits structured environments." We agree that the planning problem in their work, formulated in PDDL, is more abstract and general than our motivation and formulation, and we will make sure to reflect this distinction in the final version of the paper.
> > >
> > > It would be interesting to explore whether our approach would be directly applicable to the more general PDDL domain. The spatial assumption in our work is mainly required for dynamic replanning using D* lite, where an extension seems straightforward. For our approximate planning model, we additionally rely on access to local neighborhood structure. Whether this can be readily obtained from a PDDL formulation is not clear to us, so we leave this as an interesting direction for future work and will mention it accordingly. Thank you again for the insightful suggestion.
> > >
> > > __On "Spatial prior":__
> > >
> > > Thank you for pointing this out. The spatial prior improves performance in our setting because it reflects the structure we used to generate the experiments: the visibilities of the fields were sampled with spatial correlation, following the intuition that nearby fields are more likely to have similar values. By exploiting this structure, the inferred posteriors better align with the true underlying values, as expected in Bayesian inference.

---

> > > > ### Comment · Reviewer_cJ9W · 2025-08-04
> > > >
> > > > Thanks! This is helpful and I don't have any further questions.

---

### Official Review · Reviewer_Sttp · 2025-06-29

**Clarity:** 3
**Significance:** 2
**Originality:** 3
**Rating:** 4
**Confidence:** 4

**Summary:**

This paper presents a Bayesian inference approach to estimate an agent’s latent knowledge about the environment based on observed navigation trajectories. The approach integrates partial environmental knowledge (known state transitions) into an approximate navigation model, leveraging Bayesian inference and dynamic graph algorithms (specifically D* lite) for computational efficiency. The authors validate their approach through both simulated data and human-generated data.

**Questions:**

1. Can the authors clarify situations or conditions under which the proposed Bayesian inference method provides significant practical usability improvements over simpler models, such as the purely optimistic approach?
2. Can the authors clarify the precise meaning of "low-dimensionality" as used throughout the manuscript? Providing a threshold or clearer contextual understanding would improve clarity and traceability.
3. Given the similarities to graph learning, can the authors explicitly differentiate why their Bayesian inference approach is preferable or more beneficial compared to standard graph learning techniques?
4. In section 2.2, BAMDPs are described with uncertainty limited to transitions. Could the authors reword this section to reflect that BAMDPs more generally encapsulate uncertainty over model parameters, explicitly stating the focus here on transition uncertainty?
5. The assumption that states become "known" upon visitation resembles incremental state revelation in MDPs. Could the authors clarify if "knowledge" specifically refers to edges/transitions and explain the rationale behind this choice?
6. Why was KL divergence specifically chosen as the metric for comparing policy similarity in the parking evaluation?

**Ethical Concerns:**

["NO or VERY MINOR ethics concerns only"]

**Final Justification:**

The authors have addressed most of my initial concerns, and their clarifications on scope, dimensionality, and computational complexity were helpful. While I still view the contribution as modest and see it as a borderline accept, I have no strong objections to acceptance and think the strengths slightly outweigh the weaknesses.

**Limitations:**

Yes

**Paper Formatting Concerns:**

No major issues.

**Quality:**

2

**Strengths And Weaknesses:**

Strengths:
The paper introduces a novel Bayesian inference framework for inferring agents' latent knowledge in navigation tasks, providing a clear contribution to existing methodologies. The integration of Bayesian inference with dynamic graph algorithms (D* lite) notably improves computational efficiency, making the approach feasible for more scenarios. The empirical validation using both simulated and real human experimental data adds significant value, demonstrating the method's potential for practical applicability and for understanding human decision-making under uncertainty.

Weaknesses:
While the paper is clearly structured, several areas require further clarification and elaboration.
- The definition of "low-dimensionality" and the clarity around the specific types of uncertainty considered in BAMDPs need refinement.
- The problem framing could be clarified regarding its similarity to graph learning problems.
- The complexity claims lack explicit computational complexity analyses, and there is limited clarity regarding scalability limitations relative to alternative methods.
- Statistical analyses and detailed methodological justifications, particularly regarding evaluation measures (e.g., KL divergence) and visual representations, also need improvement.
- Another potential limitation is the practical usability of the method compared to simpler methods (e.g., purely optimistic approaches). Although the proposed method demonstrates improved log-likelihood compared to a random baseline, simpler methods (OPT) also show improvements. Clarifying when we might expect practical usability benefits of this Bayesian inference approach over simpler alternatives is warranted.

Other comments:
The paper would greatly benefit from ensuring figures are introduced and referenced in a logical and timely manner within the text, as early introductions without immediate explanations currently disrupt readability. Additionally, the rationale behind visual representations, such as connecting data points in Figures 1 and 4, should be explicitly stated or reconsidered in favor of clearer visualizations like box or violin plots. Referencing statistical analyses from the appendix within the main text, particularly when discussing results that rely heavily on distributional comparisons, would substantially enhance clarity and interpretability. The labeling in Figure 3 is somewhat unclear, making it difficult to immediately interpret. Additionally, the caption currently includes extensive detail that could be more effectively integrated into the main body of the text to improve clarity and readability.

---

> ### Author Rebuttal · Authors · 2025-07-30
>
> We thank the reviewer for the thoughtful and constructive feedback, as well as for the positive evaluation of our contribution. Below we address each of the reviewer's points and hope these clarify the reviewer's questions.
>
> ## Clarification of "low-dimensionality"
>
> In our context, "low-dimensionality" refers to settings where the number of latent knowledge configurations (i.e., belief states) is small enough to allow for exact Bayesian inference. Although we cannot define a universal threshold, in practice this typically means fewer than ~5 states in the underlying MDP. For example, in Bayesian inverse planning (ref. [5,6] in the paper), the food truck example involves 3 trucks at 3 locations, resulting in only 6 combinations = belief states. We also use this terminology to describe BEETLE. This method is applied to MDPs with up to 9 states, but it only solves the (forward) planning problem for a BAMDP, which is only a part of the knowledge inference problem. In contrast, we applied our method to problems with up to 169 states, corresponding to a latent belief space of size $2^{169}$. We will clarify these distinctions in the final version of the paper to make our use of the word "low-dimensional" more precise.
>
> ## Relation to graph learning
> As discussed in the paper, deterministic planning problems are equivalent to shortest path problems. Our approach can therefore be interpreted as learning properties about the given graph (i.e., the knowledge variables). However, our approach differs from standard graph learning, which typically reconstructs the actual environment graph from data. Instead, our goal is to reason over an agent's subjective knowledge of the graph's edges. This requires modeling both partial knowledge and adaptive planning under uncertainty. Standard graph learning approaches typically do not capture such subjective, temporally-updated knowledge states. We will use the additional page of the camera-ready version to include this relation in the final version.
>
> ## Computational complexity
> We agree that a more detailed discussion of complexity provides additional value. For exact inference, as in Bayesian inverse planning, we must consider all $2^N$ combinations of values of $K$, where $N$ is the number of states. For each configuration, computing the agent's policy and likelihood results in a complexity of $O(2^n \cdot C_\text{planning})$. Planning in the belief space using value iteration has complexity $O(|S|^2 \cdot |A| / \epsilon) = O(2^{2N} \cdot 4 / \epsilon)$, with $\epsilon$ the precision of value iteration, since the belief MDP has $2^N$ states. Additionally, planning has to be performed up to $N$ times for each configuration, as the agent is assumed to replan along the trajectory (and then all fields have been visited, including the goal). The results in a total complexity for inference of $O(N \cdot 2^{3N} / \epsilon)$.
>
> In contrast, our Gibbs-sampling-based approach scales linearly with the number of variables: we sample each $K_i$ conditioned on all others, leading to a complexity of $O(L \cdot N \cdot C_\text{planning})$, where $L$ is the number of samples. Our planning approximation avoids the complete belief space and uses the shortest path algorithm $D^*$ lite, which has worst-case complexity similar to Dijkstra, $O(|E| + |V| \log |V|) = O(4 \cdot N + N \log N) = O(N \log N)$. Similar to Bayesian inverse planning, planning needs to be done $N$ times per sample. Therefore, the total cost is $O(L \cdot N \cdot N \cdot N \log N) = O(L \cdot N^3 \cdot \log N)$. In the final version of the paper, we will complement our runtime comparison with this analysis.
>
> ## Evaluation measures and visualizations
> We used the KL divergence in the parking evaluation to quantify how closely different planning approximations match the optimal Bayesian planner and to measure their influence on the inference results. We could use this metric only for this problem, as it is one of the few cases where the exact Bayesian-optimal policy under epistemic uncertainty is computable. For the gridworld experiment, we instead used the log-likelihood of human trajectories under each model, as ground truth policies are not available and we are given experimental data.
>
> Regarding visualizations: the grey lines connect data points from the same trajectory. We agree that the black line connecting medians may be misleading and will remove it. We will use the additional space in the camera-ready version to expand the description on this.
>
> ## Practical usability compared to simpler methods
> This may reflect a misunderstanding. The only existing simpler method is to do exact planning and/or inference (i.e., Bayesian inverse planning), which is only feasible for very small environments (such as the parking task). Alternatively, one could craft rules by hand, as we did for the baselines in the gridworld tasks, where we assumed no knowledge in case of turn-arounds. In our experiments, we use the optimistic planning model as an ablation to justify a potential cost for uncertainty, but it does not make the inference problem easier: We still need to solve the high-dimensional inference problem and find in each iteration the shortest path for solving the planning problem approximately.
>
> ## Figure placement and referencing statistical analyses
> We agree that figure placement is not optimal for reading. Due to space constraints, we had to distribute figures across pages. In the final version, due to the extra page, figures will land one page later, which should solve this issue. Additionally, as suggested, we will reference and briefly discuss the statistical analysis results from the appendix to improve clarity.
>
> ## Figure 3 labeling and caption
> As also noted by Reviewer 93XY, we will add a colorbar label to improve clarity. Our intention in using detailed figure captions was to make the paper easier to interpret through visual inspection. If clarity would be improved by integrating these details into the main text, we are happy to make that revision.
>
> ## Definition of BAMDPs in Section 2.2
> We based our definition of BAMDPs on the original work by Duff [1] and the survey by Ghavamzadeh et al. [2]. Both define BAMDPs in terms of uncertainty over transition dynamics. We agree that BAMDPs can be generalized to include uncertainty over other model parameters (e.g., rewards).
>
> ## Relations to incremental state revelation
> We agree with the reviewer that our formulation, in which states become "known" upon visitation, is indeed similar to incremental state revelation mechanisms used in exploration-focused RL algorithms such as $E^3$. We clarified this connection also in the comment to reviewer 93XY. The most important differences are that we consider the inverse problem and that, in our case, model uncertainty leads to aversion. We will make this connection clearer in the final version.
>
> ## Clarifying "knowledge" as edges/transitions
> Yes, "knowledge" in our model refers to information about the transitions (edges) in the environment. In many cognitive tasks, the goal (e.g., a target location) is explicit and known. The challenge lies in discovering or reasoning about how to reach them. Modeling knowledge over transitions allows us to infer what structural information agents incorporated into their plans. This enables us, for example, to investigate how topology influences awareness or memory in human planning. With that, our approach complements approaches like inverse RL, which aims at inferring goals. We will clarify this in the revised text.
>
> ## References
> [1] M. Duff (2002). Optimal Learning: Computational procedures for Bayes-adaptive Markov decision processes. PhD thesis, University of Massachusetts Amherst
>
> [2] M. Ghavamzadeh, S. Mannor, J. Pineau, & A. Tamar (2015). Bayesian reinforcement learning: A survey. Foundations and Trends in Machine Learning, 8(5-6), 359-483

---

> > ### Author Response · Authors · 2025-08-05
> >
> > We sincerely appreciate your thoughtful review and the time you invested in evaluating our paper. As the discussion phase comes to an end, we wanted to follow up and ask whether our rebuttal addressed all of your questions, or if any points remain unclear. We would be glad to clarify further if needed, and we truly value your input and feedback.

---

> > ### Comment · Reviewer_Sttp · 2025-08-06
> >
> > Thank you for your thorough and thoughtful responses. You have addressed my questions and concerns, and I appreciate the additional clarifications provided, especially regarding dimensionality definitions and computational complexity.
> >
> > With the proposed improvements and clarifications, my overall view of the paper has become more positive.

---

### Official Review · Reviewer_93XY · 2025-07-02

**Clarity:** 3
**Significance:** 2
**Originality:** 2
**Rating:** 4
**Confidence:** 3

**Summary:**

+ Summary & Contributions
	- This paper tackles the problem of inferring agent knowledge about the environment based on an input trajectory of agent behavior.
	- Opting to directly model whether individual states are either "known" or "unknown" explicitly, the authors employ Gibbs sampling for performing approximate Bayesian inference to determine which transitions of the environment are known to the agent given a trajectory of behavior.
	- Empirical results against a variety of competing models of human participants' planning based on partial knowledge of the transition function confirm that the proposed model is competitive at providing a plausible explanation of human planning behavior and accurate inferences about the environment.

**Questions:**

My questions for the authors should be clear from the comments above.

**Ethical Concerns:**

["NO or VERY MINOR ethics concerns only"]

**Final Justification:**

After the discussion period, I know have clarity into the exact problem the authors endeavored to solve, where an initial prior or knowledge state must be inferred based on trajectory observations where the underlying strategy employed by an agent to map initial beliefs onto the observed behavior is unknown. To address this problem via Bayesian inference, the authors posit a particular planning model and corresponding likelihood function (Section 3.2) from which they may perform posterior inference over the agents initial/prior state of knowledge about the world.

Empirical results affirm that this proposed model for inference is more reliable than a diverse collection of baseline computational models. Additional experiments also study the robustness of the model to inaccuracies in the authors' choice of underlying planning model.

While I am not in the best position to evaluate this work on the axis of originality, I suspect is that this work would score highly on this axis provided that the baseline methods are reasonably chosen. Their choices seem reasonably to my eyes and my impression from the other reviewers' comments is that this was not a point of contention. As for significance, I'll defer to other reviewers with more experience on assessing computational models for cognitive science, which is outside my area of expertise.

**Limitations:**

Yes

**Quality:**

3

**Strengths And Weaknesses:**

+ Quality
	- Strengths
		- N/A
	- Weaknesses
		* Major
			- I suspect I may be misunderstanding something fundamental about the problem the authors are trying to solve. A state $s \in \mathcal{S}$ of a finite MDP is considered known (with $\tilde{k} = 1$) if, for each possible action $a \in \mathcal{A}$, the agent knows the exact successor state $s' = \mathcal{T}(s,a)$. It seems like the exposition of Section 3.1 highlights the fact that only maintaining direct beliefs about $\mathbf{K}$ discards too much information for tractable, exact Bayesian inference thereby necessitating the use of a subsequent Gibbs sampling procedure. Alternatively, since the MDP is finite and deterministic, why not proceed in the standard way by maintaining beliefs about the full MDP transition function as a collection of $|\mathcal{S}||\mathcal{A}|$ independent Dirichlet distributions (one for each possible next-state transition distribution)? Since the transition dynamics are deterministic, a reasonable prior would, for instance, be uniform with an initial parameter $\alpha_0 \ll 1$ (or even $\alpha_0 \ll 0.1$) such that any single observation of a next-state transition for any state-action pair results in a (nearly) Dirac delta distribution with all mass concentrated on the observed next state. Then, with any prior over the MDP transition function (my understanding is that the authors already assumed access to the prior $p(\mathbf{K})$), one can obtain the exact Bayesian posterior via Dirichlet-categorical conjugacy. From this Dirichlet posterior distribution, it seems like one should be capable of inducing posterior beliefs over $\mathbf{K}$ by simply iterating through all state-action pairs and assigning $\tilde{k}_i$ appropriately for each state if all successor states for each action are known or not. If my reasoning up to this point is correct, it would seemingly obviate Section 3.1 as well as any subsequent references to challenges incurred due to the use of the Gibbs sampling procedure. Meanwhile, it seems like the resulting solution would bear some resemblance to the shortest path computation described in L177-186 while accounting for the per-transition probabilities under the Dirichlet prior/posterior.
			- The authors assume that there is some parameter $q$ used to model uncertain transitions. Presumably this parameter is in the unit interval $q \in [0,1]$, but it seems quite bizarre that it is just a single value used for all transitions. Again, maybe this is just a convenient yet crude modeling choice based on representing epistemic uncertainty in $\mathbf{K}$ directly, rather than over the MDP transition function itself.
		* Minor
			- The presentation of RL and Bayesian RL in Section 2 is somewhat lacking relative to what a machine learning paper might offer. Of course, space is always at a premium and (related to my subsequent comments) this is perhaps a deliberate choice for a paper not so focused on a machine learning audience.

+ Clarity
	- Strengths
		- The paper is clearly written and reasonably structured.
	- Weaknesses
		* Major
			- N/A
		* Minor
			- The construction blockage on the way to the airport example used in the introduction doesn't really make much sense in a fully-observable environment. An external observer receiving the sequence of states encountered by the driver/agent should presumably get to observe the unexpected construction blockage just as the agent did, giving an immediate understanding of why the detour in the airport route occcurs.
			- I'm guessing there's a typo in the notation for $\mathcal{E}^k$ (L129) since $\tilde{k}_j$ isn't defined; maybe $s'$ is supposed to be $s'_j$, although I don't think this captures the idea that the successor state for each possible action executed from $s_i$ is known.


+ Originality
	- Strengths
		- Overall, the approach taken by this paper seems novel although I am not familiar enough with the cognitive science literature and existing models of human exploration to evaluate the novelty of the work with certainty.
	- Weaknesses
		* Major
			- N/A
		* Minor
			- The exposition of L124-138 is reminiscent of the classic E^3 algorithm (Kearns & Singh, 1998) but with a Bayesian twist. It would be nice for the authors to draw that connection, which would resonate quite strongly with RL researchers who read this paper.

+ Significance
	- Strengths
		- While these results may be insightful for a cognitive science audience, there seems to be little or no insight communicated for a machine learning and, in particular, reinforcement learning (RL) audience. I list this as a strength because, based on my reading of the paper, this is the authors' intention and is meant to be seen as a feature, not a bug.
	- Weaknesses
		* Major
			- A caveat of the strength comment directly above is that, while this paper may offer excellent cognitive science contributions, it seems like a rather poor fit for a machine learning venue like NeurIPS. Presumably, this paper at NeurIPS would intend to draw in RL researchers, in which case, there is a glaring gap in the exposition for what computational RL researchers should take away from these results.
		* Minor
			- N/A


+ Final Remarks
	- While this paper scores highly on clarity and while I have no major critiques with regards to originality, I have (potentially) identified some major issues with this paper on the axis on technical quality. Putting those issues aside, the more glaring obstacle is an issue of significance; this paper reads more as a cognitive science paper than a machine learning paper, with very little contribution towards the latter. I would highly encourage the authors to resubmit this work (perhaps without any modification) to a cognitive science venue where it will have a much higher chance of being encountered by a cognitive science audience who would appreciate these contributions and findings in human experiments.

---

> ### Author Rebuttal · Authors · 2025-07-31
>
> We thank the reviewer for the detailed comments and the opportunity to clarify our contributions and modeling choices. We believe most concerns stem from a misunderstanding, and we are happy to clarify them below.
>
> ## Dirichlet beliefs for exact inference about $K$
> We appreciate the reviewer's thoughtful suggestion to use a Dirichlet prior over transitions, which is indeed the classical Bayesian RL approach for modeling planning under transition uncertainty. Our main goal, however, is not to model how the agent learns about the environment, but rather to infer what the agent already knew at the start of a given trajectory, which makes it an inverse problem.
>
> In the reviewer's suggested formulation, our goal would be to infer the prior (which we term "knowledge") about state transitions based on a given trajectory, which the reviewer proposed to set to $\alpha_0 \ll 1$. This would require inference over the space of prior distributions, making the posterior a distribution over Dirichlet parameters. This inference problem cannot be addressed by conjugacy as the likelihood non-trivially depends on the initial knowledge via the agent's policy. With this formulation, it is not even clear how the agent's policy under the Dirichlet belief can be computed, which is required for evaluating the likelihood.
>
> To avoid this complexity, we consider a simplified model: Instead of Dirichlet beliefs for each transition, we assume binary knowledge for each state: each state is either fully known or unknown (with respect to its transitions).
> In addition to the lower-dimensionality of the representation, this model is structurally advantageous for computation: First, since our belief about the agent's knowledge is modeled as Bernoulli random variables, we can evaluate the conditional distributions of one belief variable, given all the others, enabling the use of Gibbs sampling. Second, we can efficiently solve the planning problem by modeling it as a shortest path problem. Third, the combination of Gibbs sampling and shortest path formulation with dynamic shortest path algorithms ($D^*$ lite) can be leveraged to reuse intermediate results in the sampling process.
>
> In summary:
> - The prior $p(K)$ is over the agent’s initial knowledge $K$.
> - The likelihood depends on the agent’s policy given that knowledge, which involves solving a planning problem.
> - This planning problem depends non-trivially on the structure of $K$, making exact Bayesian inference intractable.
> - We use Gibbs sampling not because information has been discarded, but because evaluating the posterior over knowledge states requires marginalizing over a large space.
>
> ## Use of a single parameter $q$
> The choice of a single parameter $q$ to model the probability of traversability of an unknown transition results from the simplified modeling choice that each state is either known (and its transitions are deterministic) or unknown (and transitions succeed with probability $q$). While this is a simplification, it is sufficient for many cognitive modeling tasks where we are interested in whether a subject takes into account certain information or not. Extending the model to allow an individual value for each transition would lead to the Dirichlet model again, which is not tractable. Nevertheless, in some cases, we can lift the restriction of a single $q$ to a certain degree, for example, when using different values for different structures in the maze, such as corridors and junctions.
>
> ## Airport Example
> We appreciate the feedback on the airport example in the introduction. The purpose of the example is not to argue that the environment is partially observable, but rather to illustrate why inferring the agent's prior knowledge is crucial for interpreting behavior. The external observer has access to the complete environment, including the construction blockage. However, the agent's detour into the blocked road only makes sense in this environment if we assume that the agent did not know about the construction in advance. This illustrates the type of inference our method performs.
>
> ## Typo in line 129
> Thank you for pointing this out. Indeed, $s'$ should be $s_j$ to correctly reference the knowledge of that state. It captures the idea that either the predecessor or successor must be known to know a transition.
>
> ## Connection to $E^3$ algorithm
> We appreciate the reviewer's insight regarding the connection to the $E^3$ algorithm. Our planning model indeed shares the idea of separating known and unknown states, but there are also important differences: In $E^3$, unknown states lead to an absorbing state with maximal reward, which incentivizes exploration. In our model, in contrast, unknown states are traversable with probability $q$, while with probability $1 - q$, the agent stays in the same state. This leads to uncertainty aversion, rather than exploration. We will include this comparison in the revised version to help readers better contextualize our approach.
>
> ## Suitability for NeurIPS
> We respectfully disagree with the claim that our work is a poor fit for NeurIPS. NeurIPS has long welcomed interdisciplinary contributions that combine machine learning with cognitive science, neuroscience, and related fields. As stated in the call for papers, topics include "Neuroscience and cognitive science (e.g., neural coding, brain-computer interfaces)" alongside "Reinforcement learning (e.g., decision and control, planning, hierarchical RL, robotics)". Our work lies at the intersection of reinforcement learning, Bayesian inference, and cognitive modeling. Several recent NeurIPS papers (see references [1-4] below) tackle similar questions on models for navigation and control. We therefore believe it contributes meaningfully to the NeurIPS community.
>
> ## References
> [1] Zhi-Xuan, T., Mann, J., Silver, T., Tenenbaum, J., & Mansinghka, V. (2020). Online Bayesian goal inference for boundedly rational planning agents. Advances in neural information processing systems, 33, 19238-19250.
>
> [2] Binz, M., & Schulz, E. (2022). Modeling human exploration through resource-rational reinforcement learning. Advances in neural information processing systems, 35, 31755-31768.
>
> [3] Chandra, K., Chen, T., Li, T. M., Ragan-Kelley, J., & Tenenbaum, J. (2023). Inferring the future by imagining the past. Advances in Neural Information Processing Systems, 36, 21196-21216.
>
> [4] Damiani, F., Anzai, A., Drugowitsch, J., DeAngelis, G., & Moreno Bote, R. (2024). Stochastic optimal control and estimation with multiplicative and internal noise. Advances in Neural Information Processing Systems, 37, 123291-123327.

---

> > ### Author Response · Authors · 2025-08-05
> >
> > Thank you for your detailed feedback and the time you took to evaluate our paper. In our rebuttal, we tried to provide clarifications to better explain our work and address the concerns you raised. Since the other reviewers are leaning toward acceptance, we wanted to especially follow up with you and ask whether our response helped resolve your doubts, or if any uncertainties remain. If so, we would be grateful if you would consider revisiting your evaluation. Thank you again for your thoughtful feedback.

---

> > ### Comment · Reviewer_93XY · 2025-08-06
> >
> > I thank the authors for their time and effort in providing a rebuttal response to my review.
> >
> > I now have better understanding of what problem the authors are trying to solve; this clarity implies that the paper warrants a higher score on quality than what I initially provided in my review, and I will increment it in my final review accordingly.
> >
> > When trying to infer an agent's prior knowledge about the world given the trajectory of subsequent decisions made, it seems like the choice of likelihood (encoding assumptions about how an agent uses its epistemic uncertainty about the world to guide decision-making) seems quite important. If I'm understanding correctly, the authors' assumptions around this point are outlined in full within Section A.2.
> >
> > Why do we think Section A.2 is right or accurate (in some sense)? Assuming the answer to the previous question isn't definitive (not of the form "Section A.2 must be correct because X"), how robust is this model to the details of Section A.2 being wrong? Are there alternative hypotheses (non-softmax policies that encode alternative exploration strategies like Thompson Sampling) that could be encoded instead of Section A.2 and still plugged into this model? I realize I've asked three new questions here during the discussion period; my goal is not to unnecessarily tax the authors and so I would not hold it against them if they give short responses to my questions.
> >
> > I also agree with the authors that there is nothing wrong with this paper being oriented towards a cognitive science audience. While I struggle to see significant value here for a computational RL audience (which I am a part of), the authors shouldn't be penalized for that.

---

> > > ### Author Response · Authors · 2025-08-06
> > >
> > > We are glad that our rebuttal helped clarify our work, and we appreciate your follow-up questions.
> > >
> > > > [...] it seems like the choice of likelihood (encoding assumptions about how an agent uses its epistemic uncertainty about the world to guide decision-making) seems quite important. If I'm understanding correctly, the authors' assumptions around this point are outlined in full within Section A.2.
> > >
> > > That is correct. Section A.2 provides the formal definition of the planning model and the corresponding likelihood, which was introduced in Section 3.2. It also describes how this model can be efficiently computed within the Gibbs sampling procedure.
> > >
> > > > Why do we think Section A.2 is right or accurate (in some sense)?
> > >
> > > We motivate the model in section 3.2. It is based on the optimal Bayesian planner, with two key differences: First, to account for variability, we model a stochastic policy using a softmax over path reductions, following prior work (references [41, 38, 8, 39] in the paper). Second, for tractability, we assume planning is based on the agent's current belief only, ignoring effects of future information gain in the future. Whether these assumptions match actual human planning is an empirical question. To address this, we conducted the evaluation in section 4.1 ("Planning model"), comparing the likelihoods of different planning models on real human data.
> > >
> > > > Assuming the answer to the previous question isn't definitive (not of the form "Section A.2 must be correct because X"), how robust is this model to the details of Section A.2 being wrong?
> > >
> > > We explored this in section 4.2 using the parking problem. In the second part of this evaluation, we analyzed how inference is affected when the assumed planning model in the inference procedure differs from the model with which trajectories were generated (Figure 5D). Specifically, we evaluated how results change when trajectories are generated using the optimal Bayesian policy, but the inference method assumes a different planning model, such that the one that ignores the effects of knowledge acquisition along the path.
> > >
> > > > Are there alternative hypotheses (non-softmax policies that encode alternative exploration strategies like Thompson Sampling) that could be encoded instead of Section A.2 and still plugged into this model?
> > >
> > > This is a great question. In this work, we only considered softmax policies, as they have been commonly used for modeling variability in navigation tasks. In principle, one could use Thompson Sampling to generate trajectories by sampling a set of transitions and solving the corresponding shortest-path problem. However, using such a policy model for inference is not straightforward: computing the likelihood of a trajectory under a policy based on Thompson sampling is not directly possible, since the policy distribution is not given in closed form.

---

> > > > ### Comment · Reviewer_93XY · 2025-08-06
> > > >
> > > > I thank the authors for their prompt, detailed reply to my questions.
> > > >
> > > > Based on this discussion, I will increase my score to reflect my updated understanding of the quality and significance of this work.

---

### Official Review · Reviewer_pZq4 · 2025-07-02

**Clarity:** 4
**Significance:** 2
**Originality:** 3
**Rating:** 4
**Confidence:** 4

**Summary:**

This paper address the problem of inferring a latent decision making model from observed behaviours of another agent. The problem differs from standard inverse RL or inverse optimal control, in that the latent model may contain uncertainty about the true world model, i.e., the dynamics of some states may not be known to the observed agent. The problem is addressed by modelling each state of the underlying model as known or unknown in the latent model, and then performing Gibbs sampling to compute a posterior over whether states are known or unknown, with the likelihood being given with respect to the observed policy. The application setting is to recover navigation models from observation.

**Questions:**

How does this technique compare to conventional IRL on these experiments?
Why is the posterior so high variance on the experiment data for figure 4?

**Ethical Concerns:**

["NO or VERY MINOR ethics concerns only"]

**Final Justification:**

I read the authors response and found it helpful. I continue to be positively disposed towards this paper.

**Limitations:**

Yes.

**Quality:**

3

**Strengths And Weaknesses:**

The problem in general is interesting, the written text is clear, and I enjoyed reading the paper.

The major limitation of this paper is that it is not really clear what the purpose of the result is. I understand the overall problem of recovering a latent model to model behaviour, and I like the problem setting that recognises that the acting agent may in fact not have a complete model of the world. However, the restriction to a discrete MDP, and the assumption that the *only* thing the acting agent does not know is the transition dynamics both strike me as relatively artificial, and poorly aligned with inferring real navigation models. (Inferring latent navigation models is actually a very active area of research in spatial psychology. Reading about how traffic engineers recover navigation models is super interesting!) The provided experimental results show the technique works, but are not especially compelling motivation. The underlying techniques (inference by sampling with likelihood computed from the observed trajectory) seems relatively straightforward, so without a clearer connection to a real problem, it's hard to know what the contribution is.
I was also surprised not to see a comparison to conventional IRL --- this work becomes even more interesting if there is a setting where IRL learns the wrong thing, which leads to poor performance, but the BAMDP setting learns the correct concept.

The authors list of limitations is fairly good, but these limitations are not trivial, and make me wonder even more if the proposed approach is actual useful. I had similar concerns about scalability. The authors propose using additional information sources such as eye tracking, which I completely agree with as an important step. I think that human navigation models are much more based on perceptual features than on a latent MDP where the human is not sure about the transition dynamics of some of the states.

The experimental results are reasonable, but the figures are extremely difficult to parse, partly because there are some key features that are not described --- it took quite a while to work out for figure 3 that the more blue a square is, the more likely it is that the human did not know it. I am also surprised at how high variance the posterior is for the experiment data in figure 4 -- the baseline seemed to outperform the BAMDP technique for some worlds.

I am generally inclined to support this paper, but the absence of a connection to a real application or a deep algorithm advance, the contribution of this paper is modest at best.

---

> ### Author Rebuttal · Authors · 2025-07-31
>
> We thank the reviewer for their thoughtful and encouraging review. Below, we address the main concerns in detail.
>
> ## Purpose and applicability of the results
> We appreciate this opportunity to clarify the motivation and intended use of our method. Our work is situated in computational cognitive science, where the central goal is to explain observed behavior through computational models, and not just predict it. In many controlled navigation tasks, the goal is known, but the structure of the environment is either unknown, must be memorized, or is too complex for precise planning. As researchers interested in human planning, we would like to decode the information of the environment that is relevant for the specifically followed trajectory of a subject. With this, we can gain insights about transferability (which information did subjects transfer between tasks), spatial memory (which parts of the environment were present in the subject's memory), or awareness of certain parts of the environment when planning. Our method serves as a tool to infer precisely this information. As a concrete example, we are actively using this inference framework in ongoing work to analyze how subjects' awareness depends on topological features such as corridors or junctions when solving a maze under time pressure. Further, the method could be concretely used for investigating spatial memory by investigating which parts of a larger maze a subject remembered based on its local structure.
>
> ## Generality and scalability
> We acknowledge that our approach is scoped to discrete MDPs with uncertainty over transitions. However, to the best of our knowledge, this is the first approach that enables inference over high-dimensional knowledge states from behavioral data. In our experiments, we applied the method to a task with 169 binary latent variables, which corresponds to a belief space of size $2^{169}$. This is significantly beyond what existing knowledge inference techniques, such as Bayesian inverse planning, can handle. As mentioned above, this scalability allows practical application in controlled behavioral experiments. Smaller continuous domains could also be addressed via discretization. While scaling to more complex continuous environments remains an open challenge, we believe our work lays important groundwork for doing so.
>
> ## Comparison to IRL
> We did not include a direct comparison to IRL because our method and IRL infer fundamentally different quantities: IRL estimates preferences (e.g., a cost function), while our approach infers latent knowledge (i.e., beliefs about world structure). These are not directly comparable, especially since beliefs and rewards live in different spaces. That said, in terms of trajectory likelihoods ("performance"), we expect that IRL fits the observed trajectories equally well by assigning high rewards to visited states (e.g., when the agent turns). However, most would agree that this explanation would be less appropriate, as similarly illustrated by the airport example in our introduction. If the reviewer feels it would be helpful, we are happy to include a learned cost function plot via IRL in the appendix of the final version to further illustrate this point.
>
> ## Clarification of Figure 3
> Thank you for pointing out the missing label for the colorbar in Figure 3. We will add it in the revised version to improve clarity.
>
> ## Posterior Variance in Figure 4
> We already briefly discussed the high variance of likelihoods in the evaluation and future work section. The variance arises from the behavioral variability in the human experiment, which could be caused by inattentiveness, exploratory behavior, or higher-level learning. In Figure 13 (last three columns) in the appendix, we show representative examples of the collected human data. Many participants especially in worlds 6 and 7 do not follow the most "obvious" or direct path. This could be explained by the experience that these paths were oftentiimes blocked in previous mazes, so they avoided them. To minimize such effects, we included a diverse set of gridworlds (e.g., world 6, where the direct path is indeed the shortest) and introductory trials. Nevertheless, we were surprised by the high variability of behavior and consider this an interesting direction for future investigation.

---

> > ### Author Response · Authors · 2025-08-05
> >
> > Thank you again for your thoughtful review of our paper. As the discussion phase nears its end, we wanted to kindly follow up and ask whether we could clarify your questions in our rebuttal, particularly regarding the motivation and applications of our method. We are grateful for your engagement and would value any further thoughts you might have.

---

> > > ### Comment · Reviewer_pZq4 · 2025-08-06
> > >
> > > Thanks! This is helpful and I don't have any further questions. I am still favourably disposed towards this paper, and look forward to the follow-on results described in the authors' response.

---

### Decision · Program_Chairs · 2025-09-17

**Decision:**

Accept (poster)

**Comment:**

This paper proposes a method for inferring an agent's knowledge of environment states—distinguishing between known and "unknown" areas—from observed navigation paths. Unlike standard inverse reinforcement learning approaches, it explicitly accounts for agent uncertainty. The method employs Gibbs sampling to perform Bayesian inference over state knowledge. The paper addresses an important problem and is clearly written, with experiments that are thoughtfully designed and rigorously executed. However, reviewers initially expressed concerns regarding the perceived novelty and scope of the contribution, suggesting that the work might appear somewhat narrow or incremental relative to the existing literature. Initial questions about the model motivation and spatial prior justification were effectively addressed in the rebuttal, leading to a more favorable overall evaluation of the paper.